# Staphylococcus aureus produces pain through pore-forming toxins and neuronal TRPV1 that is silenced by QX-314

Kimbria J. Blake[1], Pankaj Baral[1], Tiphaine Voisin[1], Ashira Lubkin[2], Felipe Almeida Pinho-Ribeiro[1], Kelsey L. Adams[1], David P. Roberson[3,4], Yuxin C. Ma[1], Michael Otto [5], Clifford J. Woolf[3,4], Victor J. Torres [2] & Isaac M. Chiu [1]

The hallmark of many bacterial infections is pain. The underlying mechanisms of pain during live pathogen invasion are not well understood. Here, we elucidate key molecular mechanisms of pain produced during live methicillin-resistant *Staphylococcus aureus* (MRSA) infection. We show that spontaneous pain is dependent on the virulence determinant *agr* and bacterial pore-forming toxins (PFTs). The cation channel, TRPV1, mediated heat hyperalgesia as a distinct pain modality. Three classes of PFTs—alpha-hemolysin (Hla), phenol-soluble modulins (PSMs), and the leukocidin HlgAB—directly induced neuronal firing and produced spontaneous pain. From these mechanisms, we hypothesized that pores formed in neurons would allow entry of the membrane-impermeable sodium channel blocker QX-314 into nociceptors to silence pain during infection. QX-314 induced immediate and long-lasting blockade of pain caused by MRSA infection, significantly more than lidocaine or ibuprofen, two widely used clinical analgesic treatments.

[1] Department of Microbiology and Immunobiology, Division of Immunology, Harvard Medical School, Boston, MA 02115, USA. [2] Department of Microbiology, New York University School of Medicine, New York, NY 10016, USA. [3] Department of Neurobiology, Harvard Medical School, Boston, MA 02115, USA. [4] F.M. Kirby Neurobiology Center, Boston Children's Hospital, Boston, MA 02155, USA. [5] Pathogen Molecular Genetics Section, Laboratory of Bacteriology, National Institute of Allergy and Infectious Disease, National Institutes of Health, Bethesda, MD 20814, USA. Correspondence and requests for materials should be addressed to I.M.C. (email: isaac_chiu@hms.harvard.edu)

Pain is an unpleasant sensation that serves as a critical protective response for organisms to avoid danger. Chronic pain, by contrast, is a maladaptive response of the nervous system to inflammation or injury. Given the current opioid epidemic, there is a need to better understand the molecular mechanisms of inflammatory and neuropathic pain. The mechanisms of pain during live pathogenic invasion and bacterial infection are not well understood. There are also few strategies specifically targeting pain produced by pathogens.

Nociceptors are specialized peripheral sensory neurons that mediate pain[1,2]. Nociceptors express specific molecular sensors for noxious/harmful stimuli at their peripheral nerve terminals, including transient receptor potential (TRP) ion channels that detect noxious heat, cold, protons, inflammatory lipids, and reactive chemicals[1,3]. Nociceptor cell bodies reside within the dorsal root ganglia (DRG), which propagate action potentials from the periphery to the dorsal horn of the spinal cord via their nerve central terminals to be interpreted as pain. Spontaneous, nocifensive pain reflexes are generated when nociceptors detect intense noxious stimuli, causing an immediate protective withdrawal response from the source of danger[1]. Hyperalgesia, which is the heightened sensitivity to noxious stimuli, is produced by nociceptor sensitization during inflammation or injury[1]. Pain triggers neural adaptations, such as behavioral avoidance of damaging stimuli, to allow for proper wound recovery. During infection, both spontaneous pain reflexes and hyperalgesia occur, but the underlying mechanisms of these pain modalities are unknown.

Pathogens are a major source of organismic danger and tissue damage. Bacterial, viral, and fungal infections often produce pain involving both spontaneous nocifensive reflexes and hyperalgesia[4]. Recent studies by our group and others have shown that nociceptors are capable of directly sensing bacterial ligands including cell wall components, toxins, and pathogen-associated molecular patterns[5–8]. However, these studies did not study pain during live pathogen invasion, where dynamic host–microbe interactions are at play. Thus, the specific contributions of pathogen-derived ligands to pain during infection are unclear.

In addition to needing a better understanding of the mechanisms of pain during live infection, there is a significant need to target its associated pain. Inflammation and infection is known to decrease the efficacy of local analgesics including lidocaine, by decreasing their binding to neuronal membranes and neutralization of their activity due to acidosis[9–11]. Furthermore, non-steroidal anti-inflammatory drugs (NSAIDs) can adversely affect the ability of the immune system to combat pathogens and are contraindicated for certain bacterial infections[12,13]. Therefore, there is a need to develop more effective treatments for pain that do not adversely affect host defense.

The gram-positive bacterial pathogen *Staphylococcus aureus* is a leading cause of human skin and soft-tissue infections, producing painful boils, abscesses, osteomyelitis, and cellulitis[14]. Methicillin-resistant *S. aureus* (MRSA) strains have increased in prevalence in community and hospital settings, with antibiotic resistance of growing concern, thus necessitating novel approaches to treat *S. aureus* infections. Methicillin-resistant *S. aureus* produces many virulence factors, including secreted pore-forming toxins (PFTs) of three major classes that are critical for bacterial spread and survival in the host: α-hemolysin (Hla), phenol-soluble modulins (PSMs), and bicomponent leukocidins.

In our previous studies, we determined that *S. aureus* directly activated sensory neurons, resulting in pain independent of the immune system. We found that N-formylated peptides and Hla-induced calcium influx in sensory neurons in vitro. *S. aureus* Hla mutants caused less thermal and mechanical hyperalgesia in comparison to wild-type (WT) *S. aureus*[5]. While these results lent insight into potential molecular mechanisms of pain, it was unclear how relevant they were to spontaneous pain mechanisms produced during live bacterial infection. Given that *S. aureus* produces several types of PFTs, all of which mediate virulence, the role of distinct PFTs in pain have not been investigated. We and others have also not previously developed effective pharmacological strategies to treat and alleviate pain during infection without adversely affecting host defense.

In this study, we define a role for the quorum-sensing accessory gene regulator (*agr*) system and its control of PFTs as a critical mechanism of neuronal activation during infection. We found several PFTs beyond Hla: PSMs and the leukocidin HlgAB, were each sufficient to produce pain when injected into mice. These toxins also directly induced calcium influx in neurons and robust firing of action potentials. We also developed a spontaneous pain assay utilizing live, over heat-killed bacteria, to determine the mechanisms of pain during active infection. Using this assay, we determined that spontaneous pain during MRSA infection is dependent on *agr* and Hla. In addition, we determined that the cation channel, TRPV1, mediates thermal hyperalgesia during infection, further adding to the molecular mechanisms, beyond bacterial-induced modalities, of pain during infection.

We hypothesized that QX-314, a membrane-impermeable sodium channel blocker, could be delivered into sensory neurons to alleviate pain. QX-314-silenced PFT induced neuronal activation and produced long-lasting blockade of pain caused by *S. aureus* infection without affecting bacterial elimination by the host. Therefore, we elucidate several molecular mechanisms of pain produced during *S. aureus* infection, and identify QX-314 as an effective analgesic strategy to block pain during infection.

## Results

**Live *S. aureus* produces spontaneous pain and hyperalgesia.** USA300 is a virulent community-acquired MRSA clone that is a major cause of skin and soft-tissue infections in the United States[15]. The mouse hind paw is densely innervated and often used for the study of pain reflex behaviors. To study pain during infection, we subcutaneously infected mice with different doses of USA300 into the hind paw ($5 \times 10^6$–$5 \times 10^8$ colony-forming units, CFUs) and subsequently measured spontaneous lifting/licking or flinching of the paw over 1 h. We developed this measurement assay as a representation of the sharp, spontaneous pain humans may feel during severe local bacterial infections. The doses of bacteria utilized (in CFUs) are commonly used to induce subcutaneous MRSA skin infections in mice[16]. MRSA infection induced robust and spontaneous pain behaviors within minutes (guarding/licking of the infection site) at the highest dose of USA300 ($5 \times 10^8$ CFU), but not at lower infectious doses (Fig. 1a, b and Supplementary Movie 1). Spontaneous pain peaked at 20–30 min post infection and remained sustained at a lower level up to 60 min post infection, the total time of pain analysis (Supplementary Fig. 1a). Spontaneous pain was abrogated when *S. aureus* was killed at 100 °C for 15 min prior infection, indicating a dependence on factors produced by live bacteria (Fig. 1a).

Mechanical and thermal hyperalgesia, which are heightened responses to painful stimuli, also occur during tissue injury and inflammation. *S. aureus* infection induced robust mechanical hyperalgesia, as measured using von Frey filaments, peaking 4–6 h post infection at all doses of infection tested (Fig. 1c). Mechanical hyperalgesia with lower doses of USA300 ($10^5$ and $10^6$ CFU) showed resolution to baseline by 120 h post infection, while paradoxically pain resolution occurred earlier by 24 h post infection with the highest dose ($2 \times 10^7$ CFU). *S. aureus* infection also elicited a bacterial dose-dependent induction of heat hyperalgesia as measured using the Hargreaves' radiant heat

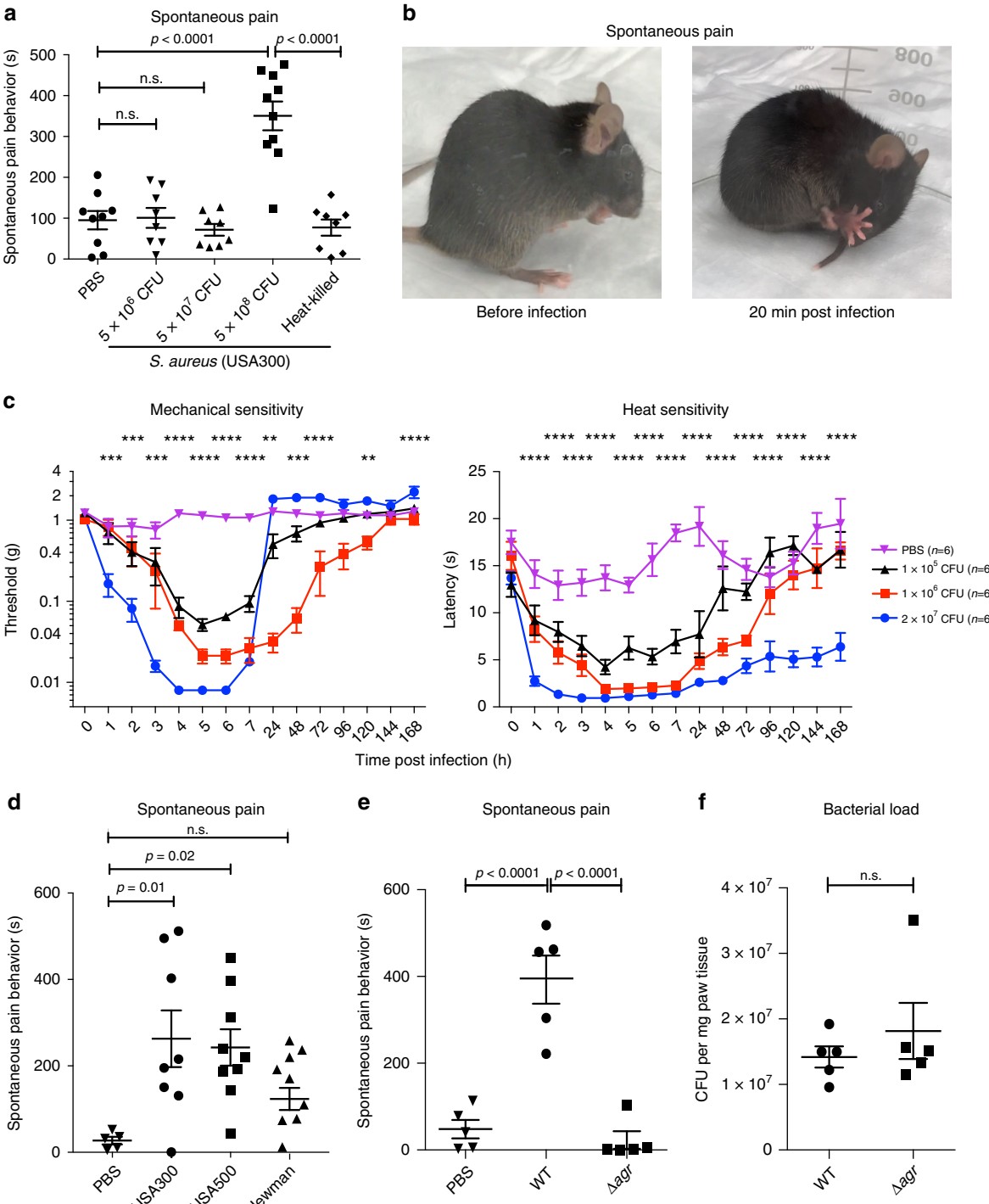

**Fig. 1** Methicillin-resistant *S. aureus* infection induces dose-dependent spontaneous pain and mechanical and heat hyperalgesia. **a** *S. aureus* infection (MRSA strain USA300) induces dose-dependent spontaneous pain reflexes (lifting/licking/flinching behaviors) in mice measured over 60 min post infection (5 × $10^6$, $n = 8$ mice per group; 5 × $10^7$, $n = 8$ mice per group; 5 × $10^8$, $n = 10$ mice per group CFU). By contrast, heat-killed bacteria (5 × $10^8$ CFU), $n = 8$ mice per group does not produce spontaneous pain. PBS control, $n = 9$ mice per group. **b** Representative images of a mouse before (left) and 20 min after infection (right) with 5 × $10^8$ CFU of *S. aureus*. **c** *S. aureus* (USA300) induces dose-dependent mechanical hyperalgesia (assayed by von Frey filaments) and heat hyperalgesia (assayed by the Hargreaves' test) measured over 168 h post infection. Two-way ANOVA with Tukey's post-tests comparing PBS vs. 2 × $10^7$ CFU *S. aureus*: **$p < 0.01$; ***$p < 0.001$; ****$p < 0.0001$. $n = 6$ mice per group. **d** Spontaneous pain induced by injection with PBS or 5 × $10^8$ CFU of different *S. aureus* strains (methicillin-resistant strains USA300 and USA500, or methicillin-sensitive strain Newman). PBS, $n = 5$; USA300, $n = 7$; USA500 and Newman, $n = 8$ mice per group. **e** Spontaneous pain reflexes induced by PBS, USA300 (WT), or USA300 isogenic mutant bacteria lacking the *agr* system (Δ*agr*). Pain depends on the presence of *agr*. $n = 5$ mice per group. **f** Bacterial load recovery from mice infected by WT or Δ*agr* USA300 1 h post infection. $n = 5$ mice per group. **a**, **d** $N = 3$ replicates; **c**, **e**, $N = 2$ replicates; **f**, $N = 1$ replicate. **a–f** Symbols represent individual mice. Statistical comparisons by one-way ANOVA with Tukey's post-tests. Error bars throughout figure, mean ± s.e.m.

assay (Fig. 1c). Heat hyperalgesia resolved to baseline sensitivity by 96 h for the lower doses ($10^5$ and $10^6$ CFU), but did not resolve for the highest dose of infection ($2 \times 10^7$ CFU), remaining at the limit of latency (~2 s) 168 h post infection (Fig. 1c). Infection-induced paw swelling and tissue damage also depended on the dose of bacterial inoculum (Supplementary Fig. 1b). To determine whether pain depended on the status of bacterial growth at the time of infection, we found infection with both mid-log and stationary phase *S. aureus*-induced similar levels of both spontaneous pain and mechanical hyperalgesia (Supplementary Fig. 2). Therefore, live *S. aureus* infection induces immediate, dose-dependent spontaneous pain, followed by robust mechanical and thermal hyperalgesia that lasts for days post infection.

**The *agr* locus mediates pain and nociceptor neuron activation.** We next compared different virulent strains of *S. aureus* in their abilities to produce pain. USA300 and USA500, two epidemic strains of MRSA[15,17], produced significant levels of spontaneous pain upon infection that were similar in magnitude to each other (Fig. 1d). The methicillin-sensitive Newman strain, which expresses lower levels of virulence determinants than USA300 or USA500[17], also produced spontaneous pain, though not significantly above PBS injection (Fig. 1d). These data indicate pain could be related to the expression of virulence factors. The bicomponent *agr* quorum-sensing system, which detects bacterial density through an auto-inducer peptide, controls the expression of *S. aureus* virulence factors including PFTs, exoproteases, and methicillin resistance genes. *agr* is activated in the transition from late-exponential to stationary phase growth, in the presence of stress, or by mammalian factors[18–20]. We found that the spontaneous pain was abrogated in mice infected with USA300 mutant for the *agr* locus (Δ*agr*), compared to WT USA300 (Fig. 1e). Mouse tissues infected with WT vs. Δ*agr S. aureus* did not differ in bacterial load recovery at the 60-min time point, indicating that the effect on spontaneous pain was not due to bacterial expansion but rather factors controlled by *agr* (Fig. 1f). Therefore, spontaneous pain reflexes produced by *S. aureus* are dependent on *agr* and correlate with bacterial virulence.

We next cultured primary DRG neurons and utilized ratiometric calcium imaging to determine whether neurons directly respond to live USA300 *S. aureus* (Fig. 2). *S. aureus* induced robust calcium flux in groups of neurons that occurred spontaneously over 15 min of co-culture (Fig. 2a, c). Many bacteria-activated neurons also responded to capsaicin, the active ingredient in chili peppers that is the prototypic ligand for TRPV1, thus marking nociceptor neurons (Fig. 2a, c). The percentage of neurons activated depended on the dosage of live bacteria, with higher concentrations of bacteria activating nearly 100% of all neurons in the imaging field (Fig. 2a, b). Neuronal activation by *S. aureus* was dependent on the *agr* virulence determinant. Significantly fewer DRG neurons responded to application of Δ*agr* mutant *S. aureus* compared to WT *S. aureus* at all bacterial concentrations tested (Fig. 2c, d). We also found that bacterial culture supernatant induced neuronal calcium flux, indicating that secreted factors can directly activate neurons (Fig. 2e, f). Moreover, supernatant from isogenic mutant USA300 lacking *agr* (Δ*agr*) produced significantly less neuronal calcium influx than WT bacteria (Fig. 2e, f). The kinetics of neuronal activation induced by live *S. aureus* matched what we observed in vivo with spontaneous pain behavior, with increasing numbers of neurons being activated over the 15-min period (Fig. 2c and Supplementary Fig. 2a). Therefore, the *agr* virulence determinant mediates both spontaneous pain produced by *S. aureus* infection in vivo and bacterial induction of neuronal calcium flux in vitro.

**Three classes of PFTs activate neurons and produce pain.** The *agr* system is a master regulator of expression of *S. aureus* PFTs that mediate bacterial virulence in host tissues[18–20]. *S. aureus* produces three classes of PFTs: (1) α-hemolysin (Hla), a beta-barrel PFT that self-oligomerizes into nanometer sized heptameric pores following membrane insertion[21,22]. (2) Bicomponent leukocidins, comprised of γ-hemolysins (HlgAB, HlgBC) and leukocidins (LukAB, LukED, LukSF (or PVL)), which require two components for assembly and bind specific receptors to oligomerize into small pores in cell membranes[23–28]. (3) PSMs, small amphipathic α-helical peptide toxins (PSMα1, α2, α3, α4, PSMβ1, β2, and δ-toxin) that are capable of lysing cells and mediating bacterial pathogenesis[29,30]. All three types of PFTs can induce cation entry into cells[31–34]. We hypothesized that these secreted PFTs could form pores in the membranes of nociceptors, thus directly depolarizing neurons to generate action potentials that produce pain.

To determine whether nociceptor neurons expressed receptors for PFTs, we analyzed our transcriptome data from FACS-purified distinct DRG neuron subsets including isolectin B4 (IB4$^+$) Nav1.8$^+$ nociceptors, IB4$^-$Nav1.8$^+$ nociceptors, and Parvalbumin (Parv$^+$) proprioceptors[35] (Supplementary Fig. 3). Analysis of mammalian host receptors for *S. aureus* PFTs showed that all neuronal subsets expressed Adam10, the receptor for α-hemolysin[21,22], and Darc, a host receptor for the bicomponent leukocidins HlgAB and LukED[27] (Supplementary Fig. 3). By contrast, Ccr5, Ccr2, Cxcr1, Cxcr2, which are other receptors for HlgAB and LukED[23,25,28], Cd11b, the receptor for LukAB[24], and C5ar, the receptor for PVL and HlgBC[25,26] are not expressed by nociceptor neurons (Supplementary Fig. 3). We previously found that Fpr2, a host receptor for PSMs[36], is expressed by nociceptors by PCR[5].

We next asked whether *S. aureus* PFTs from each class could directly induce neuronal firing using multi-electrode arrays (MEAs) to record spike discharge (Fig. 3). We focused our study on α-hemolysin (Hla), the bicomponent leukocidin HlgAB, and PSMα3, the most cytolytic of the PSMs that lyses cells in a receptor-independent manner[30,36]. Hla and HlgAB were chosen because of neuronal expression of their receptors Adam10 and Darc (Supplementary Fig. 3). We found that application of Hla to DRG neurons induced robust action potential generation as measured by MEAs (Fig. 3a and Supplementary Fig. 4a, b). Spiking activity gradually increased over 30 min of application. We next injected Hla into the mouse hind paw, and found dose-dependent induction of spontaneous pain behavior (Fig. 3b). PSMα3 also induced rapid firing of DRG neurons on MEAs, which by contrast with Hla, spiked seconds after application but decreased over time (Fig. 3c and Supplementary Fig. 4c, d). PSMα3 also caused dose-dependent action potential generation, with the most neuronal firing at high, cytolytic concentrations (μM) (Supplementary Fig. 5a). Injection of PSMα3 into mice induced dose-dependent spontaneous pain (Fig. 3d). δ-toxin (Hld), another major PSM produced by *S. aureus*, also induced dose-dependent spontaneous pain in mice (Supplementary Fig. 5c). We then tested whether HlgAB, a bicomponent leukocidin that binds DARC[27], could also activate neurons. HlgAB potently induced neuronal firing immediately upon application (Fig. 3e and Supplementary Fig. 4e, f). HlgAB also produced significant spontaneous pain in a dose-dependent manner when injected into mice (Fig. 3f). By contrast, HlgA, a single component of this toxin that cannot assemble into pores, did not produce pain (Fig. 3f).

The kinetics of pain differed between the three toxin types: whereas PSMα3 induced significant pain only in the first 5 min and then decreased afterwards, Hla and HlgAB induced progressively increased spontaneous pain post injection over 30

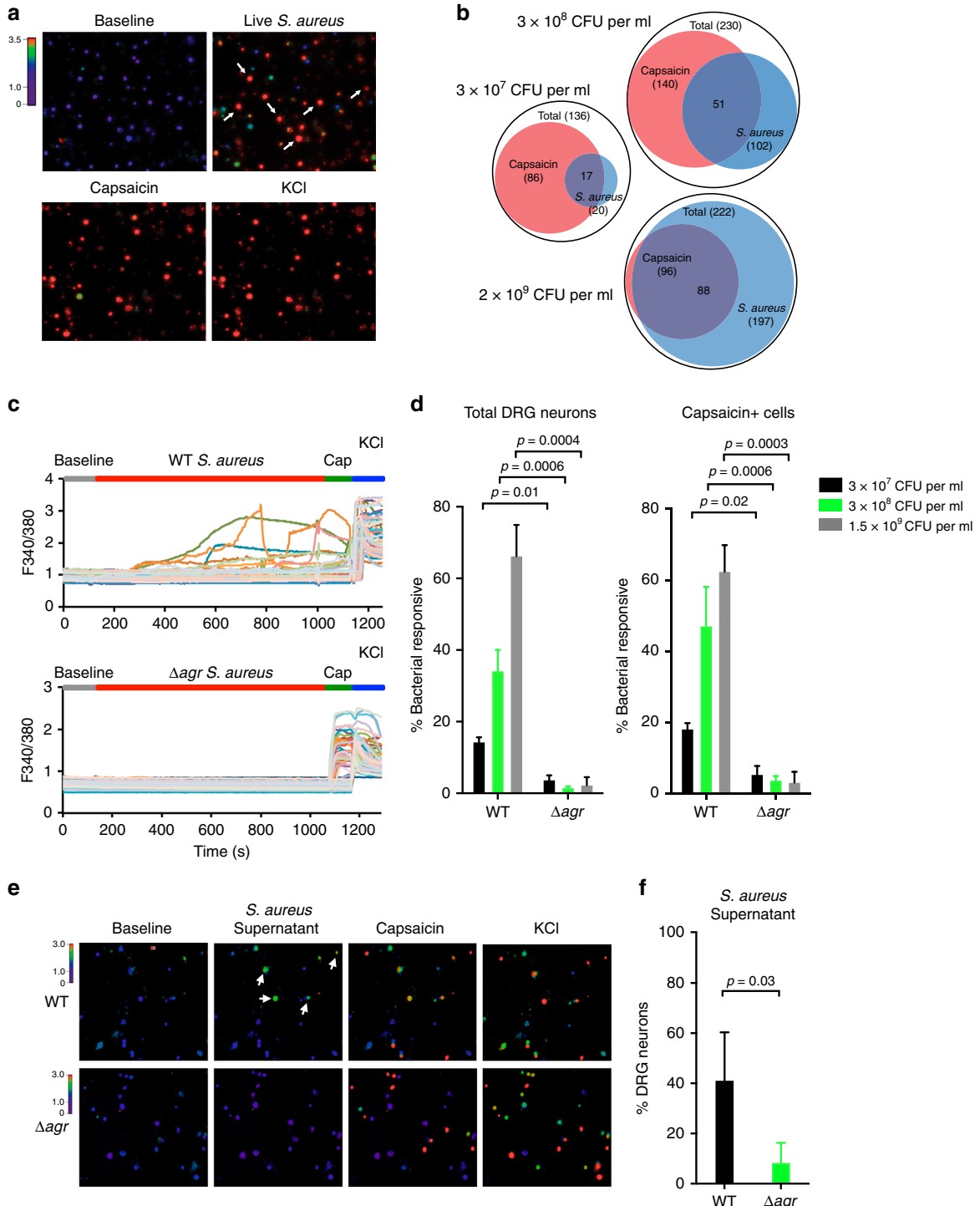

**Fig. 2** Live *S. aureus* directly induces DRG neuronal responses dependent on the *agr* virulence determinant. **a** Representative fields of Fura-2 calcium imaging of DRG sensory neurons exposed to live *S. aureus* (USA300, $2 \times 10^9$ CFU per ml), followed by capsaicin (1 µM) to activate nociceptors, and KCl (40 mM) to depolarize all sensory neurons. Arrows indicate neurons responding to bacteria. **b** Venn diagrams showing subsets of DRG neurons responding to different doses of live *S. aureus* or to the TRPV1 ligand, capsaicin. **c** Neuronal calcium traces from representative fields of neurons exposed to WT or Δ*agr S. aureus* ($1.5 \times 10^9$ CFU per ml), followed by capsaicin (1 µM), and KCl (40 mM). **d** Quantification of the proportion of total DRG neurons (left) or capsaicin + neurons (right) responding to WT or Δ*agr S. aureus* at three different bacterial doses: $3 \times 10^7$ CFU per ml: $n = 3$ fields each; $3 \times 10^8$ CFU per ml: $n = 5$ fields each; $1.5 \times 10^9$ CFU per ml: $n = 4$ fields each. *p* values, unpaired *t* test. **e** Representative imaging fields (arrows indicate neurons responding to bacterial supernatant) and **f** quantification of the proportion of neurons responding to culture supernatant from WT or Δ*agr S. aureus*. $n = 4$ fields (WT), $n = 3$ fields (Δ*agr*). **a**–**d**, $N = 3$ replicates; **f**, $N = 2$ replicates. *p* values, unpaired *t* test; error bars throughout figure, mean ± s.e.m.

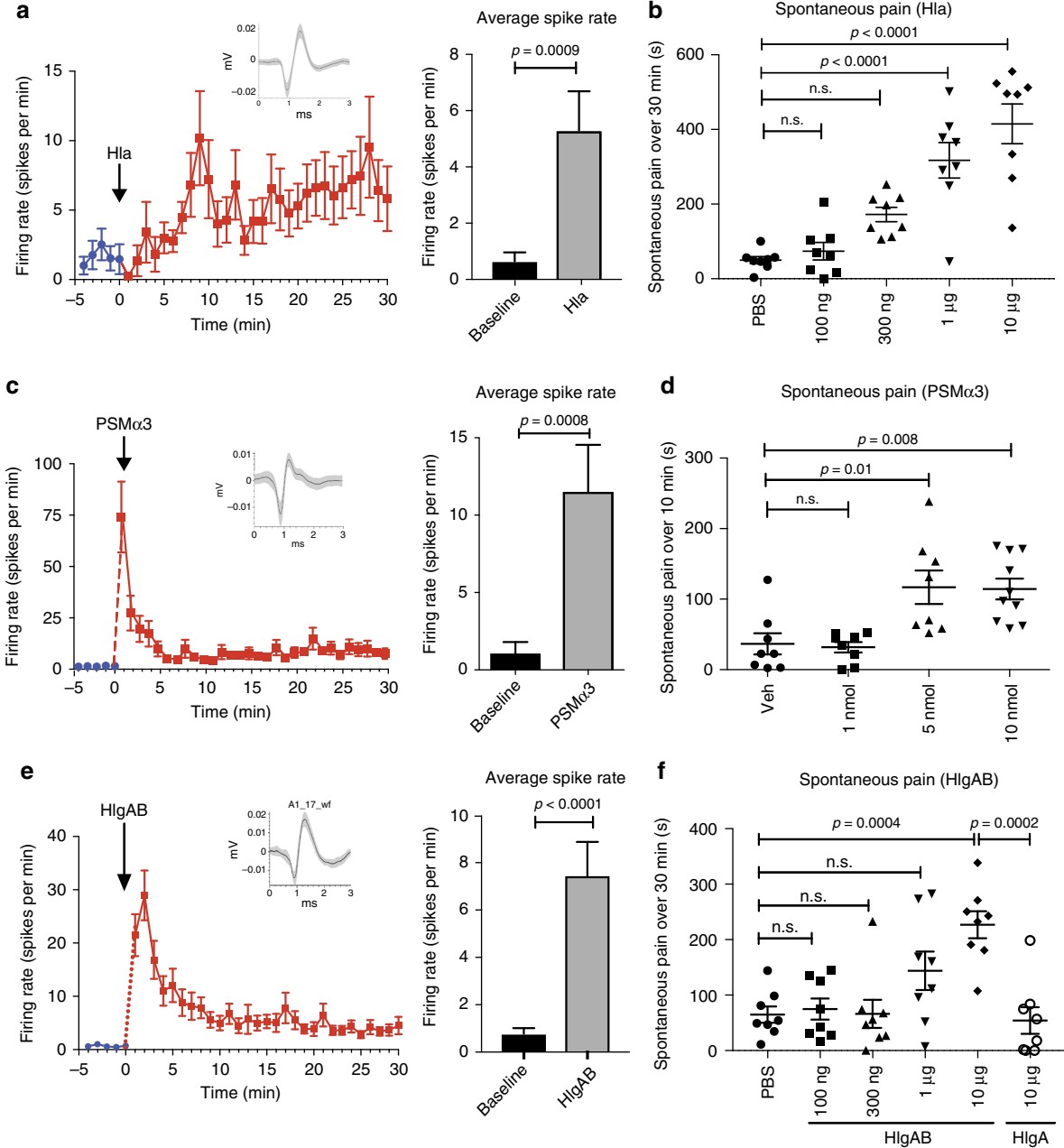

**Fig. 3** Three distinct types of PFTs from *S. aureus* induce DRG neuron firing in vitro and induce spontaneous pain reflexes in vivo. **a**, **c**, **e** DRG neuron action potential generation was quantified on multi-electrode arrays (MEAs) after application of PFTs. On left, spike rate is plotted before (blue) and after (red) application of the toxin to neurons. Arrow indicates addition of toxin. Representative action potential of an active electrode is shown above the time course. On right, average spike rate was quantified and compared at baseline (over 5 min) and after toxin addition (over 30 min) for active electrodes. **a** α-hemolysin (Hla) of 30 μg/ml (or 1 μM) induces action potential firing in DRG neurons as quantified by MEA analysis, *n* = 17 active electrodes over five plates. **b** Hla was injected into mice at increasing doses and spontaneous pain quantified over 30 min (*n* = 8 mice per group). **c** PSMα3 of 10 μM (or 270 μg/ml) induces action potential firing in DRG neurons as quantified by MEA analysis. *n* = 41 electrodes over three plates. **d** PSMα3 was injected into mice at increasing doses and spontaneous pain in mice quantified over 10 min. Vehicle is 5% DMSO in PBS. (*n* = 8 mice per group). **e** HlgAB of 3 μg/ml (1 μM of each subunit) induces action potential firing in DRG neurons as quantified by MEA analysis, *n* = 74 electrodes over seven plates. **f** HlgAB was injected into mice at increasing doses and spontaneous pain quantified over 30 min. HgAB's individual component, HlgA does not induce spontaneous pain behavior. (*n* = 8 mice per group). **a**, **c**, **e** Statistical comparisons by paired *t* test; *N* = 3 replicates per toxin. **b**, **d**, **f** Statistical comparisons and *p* values by one-way ANOVA with Tukey's post-tests; *N* = 2–3 replications per toxin. Error bars throughout figure, mean ± s.e.m.

min (Supplementary Fig. 5b). Due to the cytolytic nature of these toxins, we measured whether cytolysis occurred during neuronal recordings that could be contributing to the firing or kinetics. After 15 min of incubation of neuronal cultures with the three PFTs, as well as live *S. aureus*, the time period of our measurements (Figs. 2 and 3), we did not observe significant

lactate dehydrogenase (LDH) release (Supplementary Fig. 6), a standard cytotoxicity assay.

In summary, three distinct types of *S. aureus* PFTs (Hla, PSMα3, and HlgAB) are sufficient to rapidly generate action potential firing in neurons and to produce robust spontaneous pain within minutes upon injection into mice.

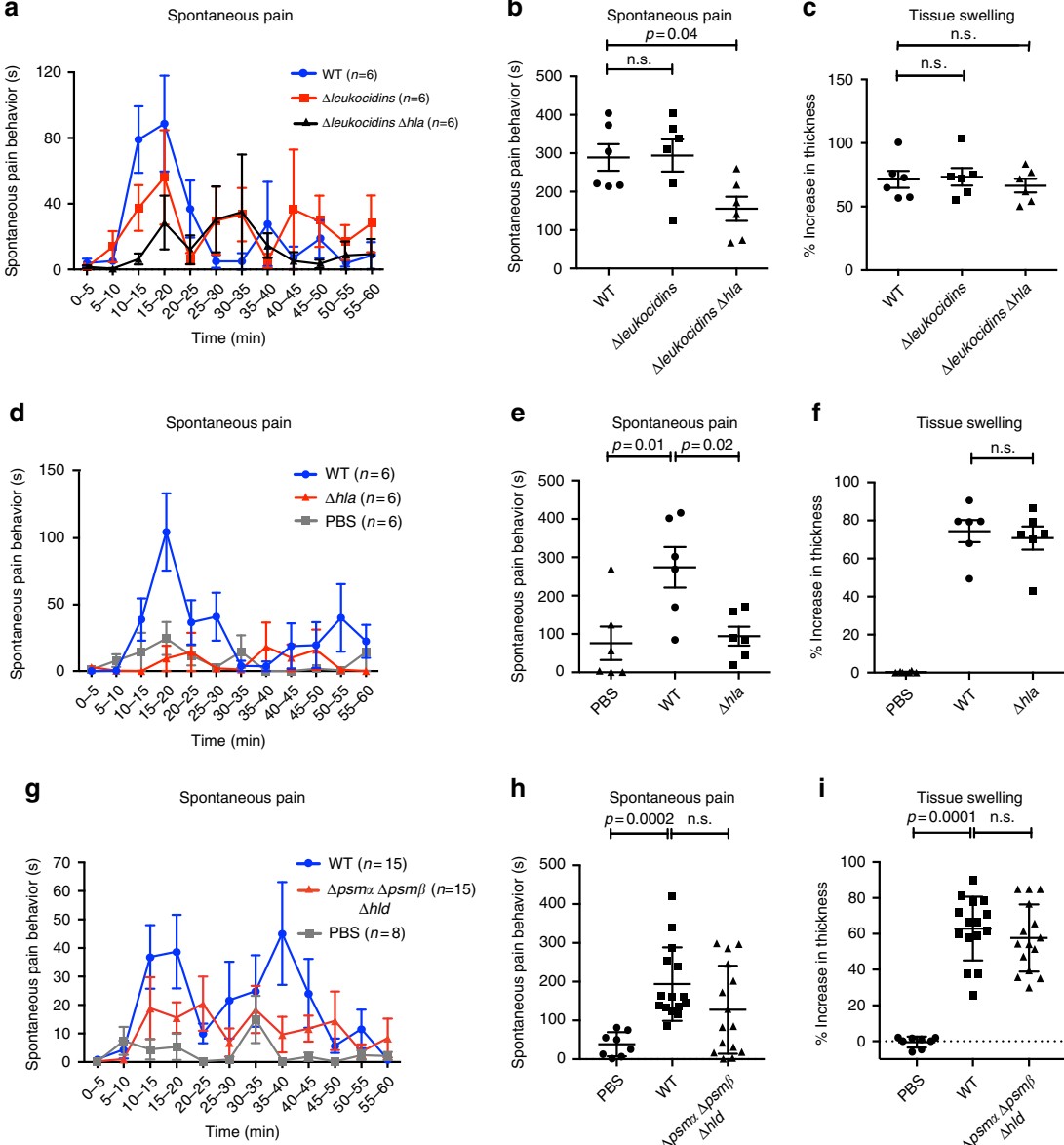

**Fig. 4** Alpha-hemolysin is necessary for spontaneous pain during live *S. aureus* infection. Mice were infected with WT or isogenic mutant strains of *S. aureus* lacking specific PFTs (USA300, $5 \times 10^8$ CFU) to determine the role of distinct toxins in spontaneous pain production. **a** Time course of spontaneous pain reflexes plotted over 5-min intervals after infection with WT *S. aureus*, Δ*leukocidins*, or Δ*leukocidins*Δ*hla* isogenic mutant *S. aureus*. $n = 6$ mice per group. **b** Quantification of pain over 60 min of infection with WT, Δ*leukocidins*, or Δ*leukocidins*Δ*hla S. aureus*. $n = 6$ mice per group. **c** Measurement of tissue swelling after infection with WT, Δ*leukocidins*Δ*hla*, or Δ*hla S. aureus*. $n = 6$ mice per group. **d** Time course of spontaneous pain behavior for WT vs. Δ*hla S. aureus*. $n = 6$ mice per group. **e** Quantification of pain over 60 min of infection with WT vs. Δ*hla S. aureus*. $n = 6$ mice per group. **f** Measurement of tissue swelling after infection with WT vs. Δ*hla S. aureus*. Hla contributes only to spontaneous pain. $n = 6$ mice per group. **g** Time course of spontaneous pain behavior for PBS vs. WT *S. aureus* vs. *S. aureus* deficient in all phenol-soluble modulins (PSMs) (Δ*psmα*Δ*psmβ*Δ*hld*). $n = 8$–15 mice per group. **h** Quantification of spontaneous pain after infection with PBS ($n = 8$ mice per group) vs. WT ($n = 14$ mice per group) vs. Δ*psmα*Δ*psmβ*Δ*hld* ($n = 15$ mice per group) *S. aureus* over 60 min. **i** Measurement of tissue swelling after infection with PBS ($n = 8$ mice per group) vs. WT *S. aureus* ($n = 14$ mice per group) vs. *S. aureus* deficient in all PSMs ($n = 15$ mice per group). **b**–**i**: *p* values, one-way ANOVA with Tukey's post-tests. **a**–**f** $N = 2$ replicates; **g**–**i** $N = 4$ replicates. Error bars throughout figure, mean ± s.e.m.

**PFTs are necessary for *S. aureus*-induced spontaneous pain.** We next wished to determine if specific PFTs played a relevant role in pain caused by live *S. aureus* infection. We infected mice with WT USA300 *S. aureus* or an isogenic mutant USA300 strain lacking all bicomponent leukocidins (Δ*leukocidins*, mutant in *hlgACB*, *lukED*, *lukAB*, *lukSF* loci), and a USA300 strain lacking both leukocidins and α-hemolysin (Δ*leukocidins*Δ*hla*). We found that while deficiency in leukocidins (Δ*leukocidins*) did not affect pain, combined deficiency in Hla and leukocidins

(Δ*leukocidins*Δ*hla*) significantly decreased spontaneous pain compared to WT bacteria (Fig. 4a, b). The degree of tissue swelling immediately following pain analysis did not differ between these strains (Fig. 4c). We next determined whether Hla was a key driver for spontaneous pain. USA300 with a single mutation in Hla (Δ*hla*) showed significantly less induction of pain compared to WT *S. aureus*-infected mice; pain in the Δ*hla* infected mice was the same level as PBS injected control mice (Fig. 4d, e). Hla was thus required for spontaneous pain

production. The degree of tissue edema following pain analysis did not differ due to Hla deficiency, indicating a dissociation of the mechanisms responsible for pain and tissue swelling (Fig. 4f). Hla deficiency also did not affect bacterial load recovery at this time point (Supplementary Fig. 7).

We next analyzed whether Hla contributed to induction of calcium flux in DRG neurons by *S. aureus*. We found that $\Delta hla$-mutant *S. aureus* induced less activation of capsaicin responsive nociceptor neurons compared to WT bacteria (Supplementary Fig. 8). However, the reduction in activation was less than what we observed with $\Delta agr$ bacteria (Fig. 2). Therefore, virulence factors controlled by the *agr* system other than Hla likely contribute to calcium influx.

We next analyzed whether PSMs played a role in pain during infection. We compared WT USA300 with isogenic mutant bacteria deficient in all PSMs ($\Delta psm\alpha\Delta psm\beta\Delta hld$). While spontaneous pain was not significantly reduced in this strain compared to WT *S. aureus* during infection ($p = 0.15$), there was a trend toward decreased pain (Fig. 4g, h). Therefore, we performed a second independent experiment with isogenic mutant USA300 at single loci for PSMs: PSMα gene locus ($\Delta psm\alpha$), PSMβ locus ($\Delta psm\beta$), or the *hld* gene ($\Delta hld$), as well as bacteria deficient in all PSM loci ($\Delta psm\alpha\Delta psm\beta\Delta hld$). In this second experiment, depletion of any individual PSM loci or of all PSMs did not significantly reduce spontaneous pain compared to WT USA300, though there was still a trend toward decreased pain with total PSM deficiency (Supplementary Fig. 9). Therefore, PSMs play a minor role in spontaneous pain production, while Hla plays a major role in this phenotype (Fig. 4e). Like leukocidins and Hla, PSMs did not contribute to tissue edema (Fig. 4i).

Overall, these data show all three classes of *agr*-dependent PFTs (Hla, leukocidins, and PSMs) are sufficient to directly induce neuronal activation and produce spontaneous pain when injected into mice (Fig. 3). However, during live bacterial infections, only Hla is necessary for the induction of spontaneous pain (Fig. 4).

**TRPV1 mediates thermal hyperalgesia in *S. aureus* infection.** We next examined the molecular mechanisms of hyperalgesia produced by *S. aureus* infection, which developed later and lasted longer than the spontaneous response. Unexpectedly, absence of *agr* ($\Delta agr$) did not affect mechanical or heat hyperalgesia during infection compared to WT bacteria (Supplementary Fig. 10). The lack of phenotype with $\Delta agr$ *S. aureus* may be due to low levels of some PFTs (over non-existent) or compensatory effects due to loss of other mediators controlled by *agr* (*agr* controls expression of ~100 factors)[18]. We next determined whether other molecular mechanisms of nociception could mediate hypersensitivity. TRPV1, an ion channel expressed by nociceptors, is activated by noxious heat and is a critical mediator of heat hyperalgesia in inflammatory pain in other settings[1,3]. We hypothesized that TRPV1 may have a role in hyperalgesia during *S. aureus* infection. We treated mice with increasing doses of resiniferatoxin (RTX), a highly potent TRPV1 agonist, which leads to loss of TRPV1-expressing nerve fibers and neurons[37]. Mice were analyzed 4 weeks later for their pain responses to *S. aureus* infection (Fig. 5a, Supplementary Fig. 11a). RTX-treated mice showed significantly decreased spontaneous pain upon bacterial infection compared to vehicle-treated littermates (Fig. 5c). RTX treatment caused complete loss of heat sensitivity at baseline. Following *S. aureus* infection, RTX-treated mice did not display drops in thermal latencies, indicating that TRPV1$^+$ neurons are critical for heat hyperalgesia during infection (Fig. 5a). Resiniferatoxin did not affect mechanical hyperalgesia, indicating other subsets of sensory neurons likely mediate this pain modality (Fig. 5,

Supplementary Fig. 11a). Next, we used mice deficient in TRPV1 ($Trpv1^{-/-}$ mice) to determine the role of the ion channel in pain production (Fig. 5b, Supplementary Fig. 11b). $Trpv1^{-/-}$ mice showed significantly less induction of heat hyperalgesia following *S. aureus* infection compared to $Trpv1^{+/+}$ or $Trpv1^{+/-}$ littermates (Fig. 5b). $Trpv1^{-/-}$ mice did not show differences in mechanical hyperalgesia or spontaneous pain production compared to control littermates (Fig. 5d, Supplementary Fig. 11b). By contrast, RTX treatment abrogated spontaneous pain and thermal hyperalgesia (Fig. 5a, c). These data show that TRPV1-expressing nociceptors mediate both spontaneous pain and thermal hyperalgesia; the TRPV1 ion channel itself is mainly necessary for heat hyperalgesia during *S. aureus* infection.

**QX-314 blocks PFT induced neuronal firing and pain.** Based on the finding that PFTs are critical mediators of pain during infection, we aimed to develop an effective strategy to target pain based on these mechanisms. QX-314 is a positively charged voltage-gated sodium channel inhibitor that is normally membrane-impermeant[38]. Because QX-314 is small enough in size, it was shown that opening of large-pore cation channels can be utilized to deliver QX-314 into nociceptors to produce long-lasting pain inhibition[38,39].

We hypothesized that bacterial-induced pain and neuronal activation could also induce large openings in neuronal membranes, allowing QX-314 delivery into nociceptors to block action potential generation to silence pain. We found that Hla and PSMα3 both caused robust firing of action potentials by DRG neurons on MEA plates (Fig. 6a, c). We then applied QX-314, which produced immediate and significant blockade of action potential firing induced by either Hla or PSMα3, suggesting entry into neurons (Fig. 6a, d).

We next determined whether QX-314 affects pain production by PFTs in vivo. Mice were injected with Hla, followed by either 2% QX-314 or PBS 15 min later. The second injection decreased pain in the first minutes likely due to mouse handling. However, we observed that the Hla→PBS group showed robust pain at later time points while the Hla→QX-314 group showed little spontaneous pain behaviors, and these differences were significant (Fig. 6e, f). Therefore, QX-314 robustly silences neuronal firing and spontaneous pain reflexes induced by *S. aureus* PFTs.

**QX-314 effectively silences pain during MRSA infection.** We next determined whether the pain produced during live USA300 infections could be inhibited by QX-314, and how this compared to the efficacy of other analgesics. Because QX-314 is a positively charged quaternary derivative of lidocaine, we compared the effectiveness of these two analgesics side by side in their abilities to treat *S. aureus*-induced pain. We injected vehicle, QX-314, or lidocaine together with *S. aureus*, and measured pain production following USA300 infection. Both lidocaine and QX-314 significantly decreased spontaneous behavior produced by *S. aureus* (Fig. 7a). We next analyzed the effects of QX-314 and lidocaine on mechanical and thermal hyperalgesia. QX-314 (2%) induced significant blockade of mechanical hypersensitivity for up to 7 h post injection (Fig. 7b). By contrast, lidocaine was completely ineffective at alleviating mechanical hyperalgesia during infection (Fig. 7b). The QX-314-induced effect was analgesic, as mechanical sensitivity was raised significantly above baseline levels up to a 6 g von Frey threshold. QX-314 also induced blockade of heat hyperalgesia (~3 h), though for a shorter timeframe than mechanical pain hypersensitivity (Fig. 7c). Lidocaine had no effect on thermal hyperalgesia during infection (Fig. 7c). $Trpv1^{-/-}$ mice did not show reduced QX-314-mediated silencing of

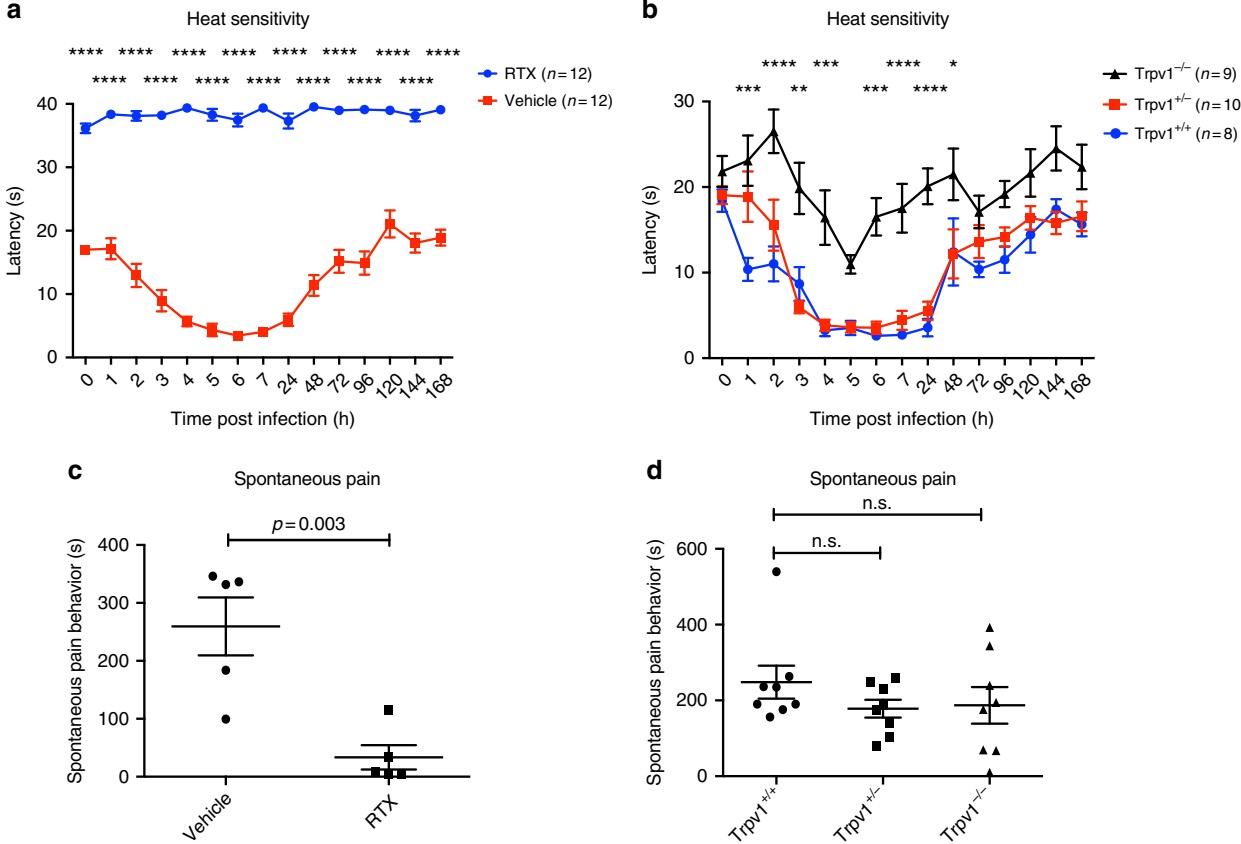

**Fig. 5** Trpv1 mediates heat hyperalgesia during *S. aureus* infection. **a** Heat hyperalgesia was measured by the Hargreaves' radiant heat test in resiniferatoxin (RTX) vs. vehicle-treated mice following *S. aureus* infection ($1 \times 10^6$ CFU, USA300). Forty seconds is the maximum cutoff for this assay. $n = 12$ mice per group. *p* values, two-way ANOVA with Sidak's post-test. **b** Heat hyperalgesia measured in Trpv1$^{-/-}$ compared to Trpv1$^{+/-}$ or Trpv1$^{+/+}$ littermates following *S. aureus* infection. Statistical comparisons shown: Trpv1$^{-/-}$ vs. Trpv1$^{+/+}$ littermates. *p* values, two-way ANOVA with Tukey's post-test. $n = 8$–10 mice per group. **c** Spontaneous pain was quantified over 60 min in RTX or vehicle-treated mice infected with *S. aureus* (USA300, $5 \times 10^8$ CFU). $n = 5$ mice per group. *p* values by unpaired *t* test. **d** Spontaneous pain was quantified over 60 min in Trpv1$^{-/-}$ mice and littermate (Trpv1$^{+/-}$, Trpv1$^{+/+}$) controls in *S. aureus*-infected mice (USA300, $5 \times 10^8$ CFU). $n = 8$ mice per group. **a**, **c** $N = 2$ replicates; **b**, **d** $N = 3$ replicates each. *p* values, one-way ANOVA, Tukey's post-test. ****$p < 0.0001$; ***$p < 0.001$; **$p < 0.01$; *$p < 0.05$. Error bars throughout figure, mean ± s.e.m.

mechanical hyperalgesia, indicating other mechanisms of entry into neurons in vivo (Supplementary Figs. 11b and 12).

Ibuprofen is a widely used NSAID that inhibits cyclo-oxygenase-mediated prostaglandin synthesis to treat inflammatory pain. However, its effectiveness in bacteria-induced pain has not been determined. Ibuprofen treatment of mice at two doses, including the maximal recommended dose for humans (40 mg/kg), was ineffective at blocking *S. aureus*-induced mechanical hyperalgesia (Fig. 7d).

We wished to determine whether QX-314 could be applied in clinical settings by treatment post infection, given the lack of efficacy for both lidocaine and ibuprofen in pain blockade. QX-314 was injected at 24 and 48 h after establishment of maximal pain hypersensitivity (Fig. 7e). QX-314 effectively produced hours-long analgesia after each injection. We also measured bacterial load recovery from QX-314 injected mice, and did not observe significant changes compared to vehicle injected mice, showing that analgesia did not adversely affect host defense against *S. aureus* (Fig. 7f). These data indicate that QX-314 is an effective approach to treat infection-induced pain.

## Discussion

Pain is a hallmark of many bacterial infections, including skin abscesses, dental carries, and urinary tract infections. However, few studies have determined the molecular mechanisms of pain

during live pathogen invasion. Our results show that several types of bacterial PFTs can directly induce neuronal calcium influx and action potential firing to produce pain. Given their prevalence in bacterial pathogens, these toxins could be a basic mechanism of pain caused during bacterial infections. Furthermore, we find that the charged analgesic QX-314 immediately silences neuronal activity caused by injection of purified PFTs, and potently blocks all major spontaneous and chronic pain modalities during live MRSA infection.

There is a great need to develop better treatments for pain during infection. Local analgesics including lidocaine and mepivacaine are neutralized by infection and inflammation[9–11]. In our study, we found that lidocaine had no effect on MRSA-induced mechanical or heat hyperalgesia. By contrast, QX-314 produced both immediate and long-lasting blockade of both pain modalities. NSAIDs, including ibuprofen, are also widely used in inflammatory pain blockade. However, our study shows that ibuprofen, even at the maximum recommended dose (40 mg/kg), has no effect on *S. aureus*-induced pain.

Mice are commonly used to study bacterial pathogenesis of several types of MRSA infections (e.g., skin, lung, bacteremia). Here, we used a subcutaneous MRSA skin infection model to assay infection-related pain, representative of cellulitis or abscess formation in humans. Human clinical MRSA isolates (such as USA300 used here) and not mouse-adapted strains are typically

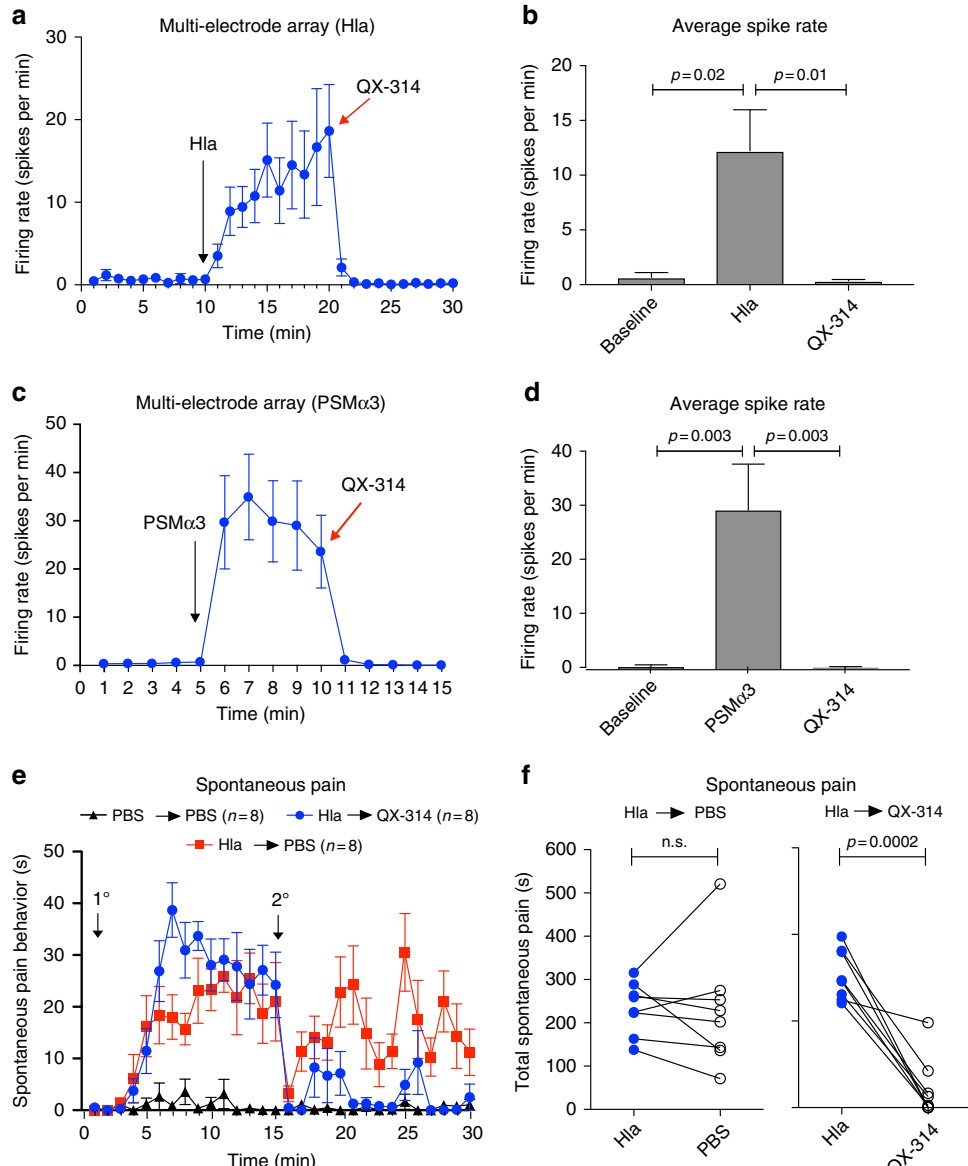

**Fig. 6** QX-314 blocks PFT induced DRG neuronal firing in vitro and spontaneous pain in vivo. **a**, **c** DRG neuronal firing measurement on multi-electrode array (MEA) plates after sequential applications of Hla (30 μg/ml or 1 μM) at 10 min and 5 mM QX-314 at 20 min (**a**) or PSMα3 (270 μg/ml or 10 μM) at 5 min and 5 mM QX-314 at 10 min (**c**). Arrows indicate time of Hla, PSMα3, and QX-314 applications; n = 20 electrodes over six plates (**a**) and n = 46 electrodes over three plates (**c**). **b**, **d** Average spike rate calculated over 5 min at baseline and after applications of the toxin (Hla (**b**) and PSMα3 (**d**)) and after application of QX-314, statistical comparisons by repeated measures (RM) one-way ANOVA with Tukey's post-tests. **e** Spontaneous pain was measured in 1-min time intervals after injection of either Hla (1 μg or 1.7 μM) or PBS into the hind paw. At the 15-min time point, mice were then injected with either 2% QX-314 or PBS (arrows indicate times of injection of each item; n = 8 mice per group). **f** Quantification of spontaneous pain over 30 min. Data in **e** shows a significant decrease in total Hla-induced spontaneous pain after QX-314 but not PBS treatment. **a**–**f** N = 3 replicates. p values, paired t tests. n = 8 mice per group. Error bars throughout figure, mean ± s.e.m.

used for these studies. Therefore, large amounts of bacteria are commonly needed to induce skin infections ($1 \times 10^7$–$1 \times 10^9$ CFU) in immunocompetent mice[16], whereas in humans a smaller inoculum could lead to significant infection. The growth and number of bacteria used in our pain assays are consistent with methods used in other *S. aureus* skin infection studies[16,30,40]. There are caveats to using mouse models of infection, including species-specific differences in receptors for leukotoxins (e.g., C5a receptor does not bind PVL in mice), and the irrelevance of secreted *S. aureus* factors that act specifically to neutralize human innate defenses[16]. Despite these caveats, the pain mechanisms we determined in this study are likely still relevant to those during human infection. While our study focuses on acute MRSA skin

infections, we hypothesize that chronic infections of the skin (e.g., atopic dermatitis) in humans may have similar mechanisms of neuronal activation; this will need to be ascertained in future studies.

Our observation of spontaneous pain reflexes produced by live bacterial infection is likely representative of the sharp, stabbing pain experienced by patients during infection[4,14], notably during the peak of infection when bacterial load is highest. To observe this phenotype, we used a large dose of MRSA ($5 \times 10^8$ CFU), albeit still in the range of bacteria used in mouse skin infection models[16]. We hypothesized that a large number of bacteria locally concentrated would allow for toxin concentrations similar to those at the peak of invasion to occur within our assay period (60

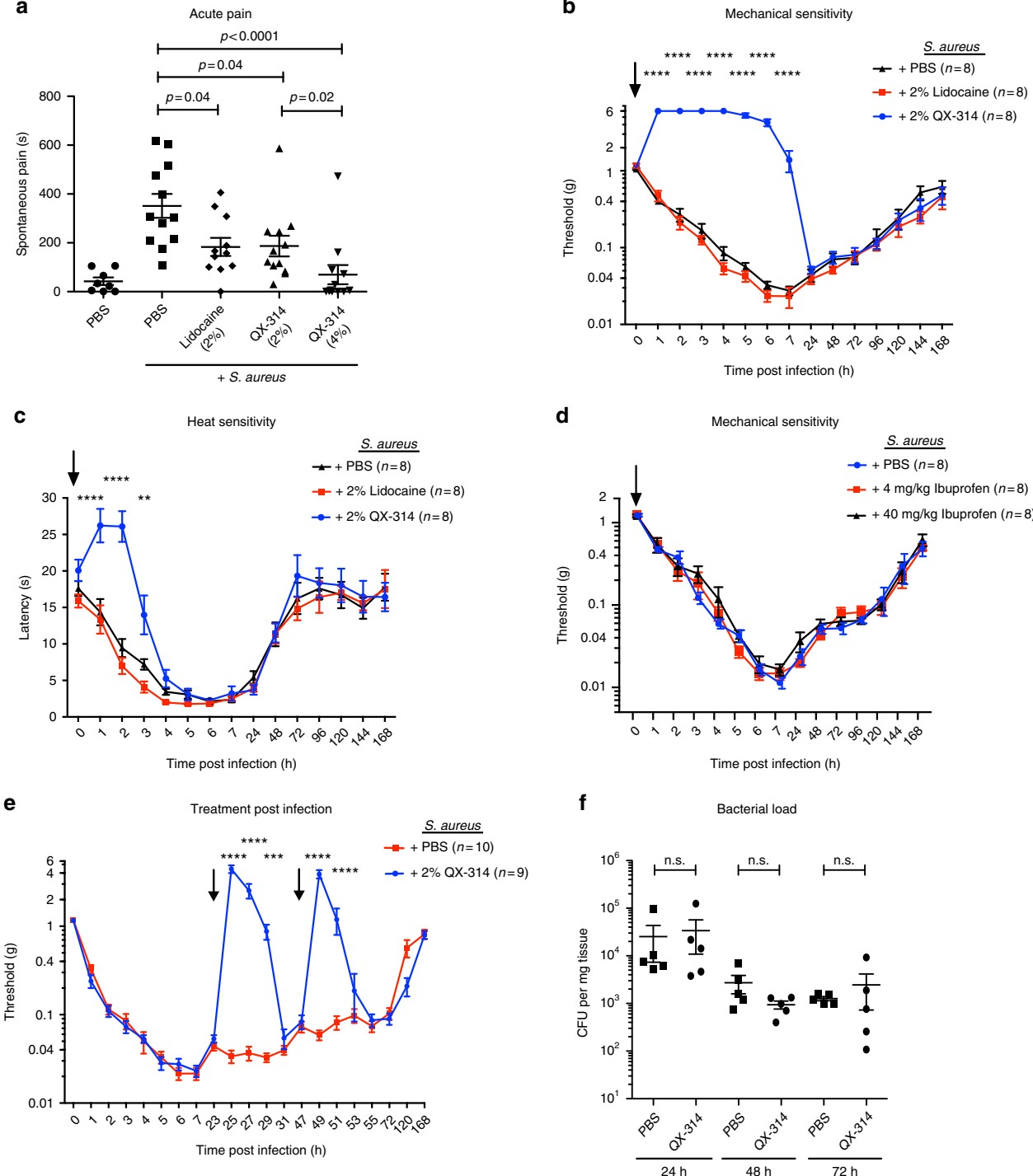

**Fig. 7** QX-314 alleviates spontaneous pain, mechanical, and thermal hyperalgesia during *S. aureus* infection. **a** Total spontaneous pain over 60 min induced by *S. aureus* ($5 \times 10^8$ CFU, USA300) injection together with QX-314 (2 or 4%), $n = 12$ mice per group; lidocaine (2%), $n = 11$ mice per group; or vehicle (PBS), $n = 12$ mice per group. Vehicle control (PBS followed with PBS treatment), $n = 8$ mice per group. $p$ values by one-way ANOVA, Tukey's post-tests. **b** Mechanical hyperalgesia induced by *S. aureus* infection ($1 \times 10^6$ CFU) was measured by von Frey hair tests. Mice were co-injected with QX-314 (2%), lidocaine (2%), or PBS at infection (arrows). Statistical comparisons: QX-314 vs. PBS, two-way ANOVA with Tukey's post-tests. $n = 8$ mice per group. **c** Heat hyperalgesia induced by *S. aureus* ($1 \times 10^6$ CFU) was measured by the Hargreaves' radiant heat test. Mice were co-injected with QX-314 (2%), lidocaine (2%), or PBS at infection (arrows). Statistical comparison: QX-314 vs. PBS. $n = 8$ mice per group. $p$ values, two-way ANOVA, Tukey's post-tests. **d** Mechanical hyperalgesia induced by *S. aureus* infection ($1 \times 10^6$ CFU) was measured in presence of ibuprofen. Ibuprofen (4 mg/kg or 40 mg/kg) or PBS was co-injected of at the time of *S. aureus* infection ($1 \times 10^6$ CFU) (arrows), $n = 8$ mice per group. $p$ values, two-way ANOVA with Tukey's post-tests. **e** Mice were infected with *S. aureus* ($1 \times 10^6$ CFU) and injected with QX-314 (2%) or with PBS at two indicated time points post infection (arrows indicate QX-314 or PBS injections). $n = 9$–10 mice per group. $p$ values, two-way ANOVA, Sidak's post-tests. **f** Bacterial load of *S. aureus* infection ($1 \times 10^6$ CFU) after treatment (1, 2, or 3 times) with 2% QX-314. $n = 5$ mice per group. **a** $N = 4$ replicates; **b**, **c** $N = 2$ replicates; **d**–**f** $N = 1$ replicate. $p$ values, one-way ANOVA with Tukey's post-tests. ****$p < 0.0001$, ***$p < 0.001$, **$p < 0.01$, *$p < 0.05$. Error bars throughout figure, mean ± s.e.m.

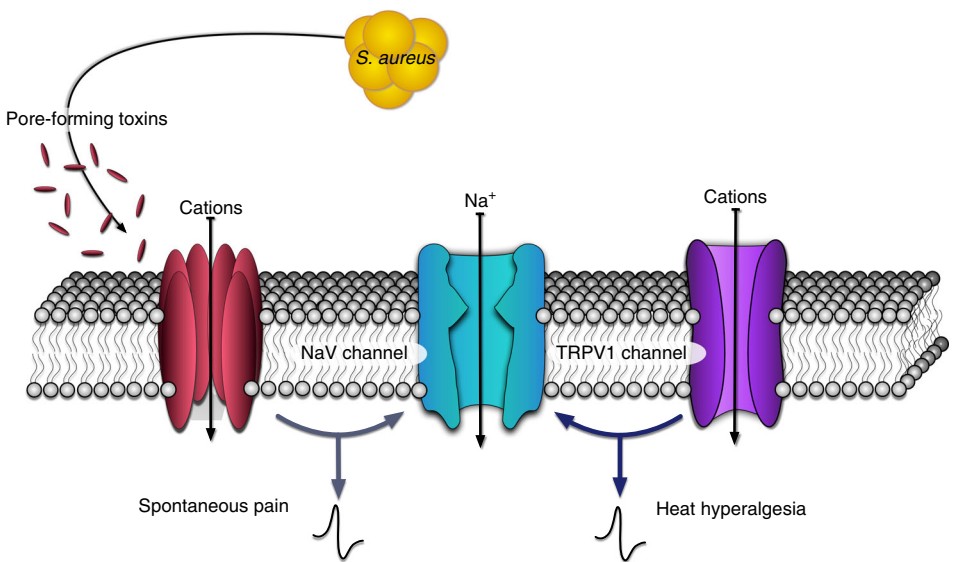

**Fig. 8** Molecular mechanisms of pain during live *S. aureus* infection. *S. aureus* induces significant spontaneous pain mediated by PFTs. *S. aureus* secretes several types of PFTs including α-hemolysin, PSMα3, and HlgAB, which can form pores in DRG neuronal membranes sufficient for cation influx and action potential generation. All three types of PFTs produce spontaneous pain when injected into mice, but only α-hemolysin is necessary for *S. aureus*-induced spontaneous pain. As a separate pain modality, *S. aureus* induces significant heat hyperalgesia, which is dependent on TRPV1 ion channels

min). In addition, as spontaneous pain occurred within 15 min, the mice are naive to this pain (i.e., they are not exhibiting pain avoidance behaviors such as sleeping, or hiding their paw[41,42]), allowing for more consistent results across animals. These avoidance behaviors are likely causative of the drop in spontaneous pain after 30 min, as we consistently observed mice sleeping intermittently after this period. Although there are caveats, we believe this assay allowed us to determine the role of key bacterial factors in mediating spontaneous pain during infection.

The *agr* quorum-sensing system, a virulence determinant that controls the expression of PFTs[18–20], was critical for spontaneous pain during infection, fitting with the hypothesis that higher production of virulence factors correlates with pain. We found that three types of *S. aureus* PFTs, including Hla, HlgAB, and PSMs, directly induced neuronal firing and pain (Fig. 8). Hla and HlgAB are secreted first as monomers, which dock in membranes and oligomerize to form pores, allowing cation influx into several mammalian cell-types[32]. Phenol-soluble modulins are peptide PFTs that also induce cation influx[31], though structures of PSM-generated pores have not been fully elucidated (only peptide nuclear magnetic resonance (NMR) structures)[43,44]. In response to PFTs, mammalian host cells turn on autophagy and other repair mechanisms[45]. Due to membrane repair, pores could be transient in nature, allowing some cationic entry before closing; these processes could account for differences in firing kinetics induced by distinct *S. aureus* PFTs. Given the cytolytic nature of PFTs, it would also be interesting to determine if infection induces permanent damage to nociceptor nerve terminals or loss of neurons during infection that would result in long-term pain phenotypes.

The complex interplay of PFT expression by *S. aureus* could also contribute to pain phenotypes. Berube et al.[46] found that USA300 deficient in PSMs (particularly PSMα and Hld mutants) showed reduced Hla production compared to WT, bacteria at 3 h in culture, though this Hla production was restored by 6 h. Our bacterial inoculums are likely between the two time points of their study. We observed a non-significant trend toward decreased spontaneous pain in PSM deficient strains. Thus, this phenotype

could be explained by decreased Hla production within USA300 PSM mutants, rather than the absence of PSMs.

Our study shows that distinct pain modalities occur during live MRSA infection—spontaneous pain, thermal, and mechanical hyperalgesia. We found that the TRPV1 ion channel mediated heat hyperalgesia, but not spontaneous pain reflexes, during *S. aureus* infection (Fig. 8). TRPV1 detects noxious heat, capsaicin, and protons ($H^+$), playing a major role in thermal hyperalgesia[3]. TRPV1 could be sensitized during infection through several mechanisms that require further study. Bacterial infections induce acidosis, and protons could directly gate TRPV1. Another potential mechanism is cytokine-mediated sensitization of TRPV1 through phosphorylation cascades. Other potential mechanisms of hyperalgesia include the action of bacterial proteases, oxidative mediators, and cytokines released by immune cells during inflammation. Equally likely is the involvement of other ion channels or receptors we have not yet considered.

We found that QX-314 potently silences both *S. aureus*-induced spontaneous pain and hyperalgesia. QX-314 is a positively charged sodium channel blocker that is normally membrane-impermeant. Previously, TRPV1 and TRPA1 were shown to allow the delivery of QX-314 into nociceptors through the transient pores formed by the opening of these cation channels[38]. Recently, Ji and colleagues showed flagellin, a component of bacteria activates A-fiber neurons, and that, co-administration of flagellin with QX-314 could silence neuropathic pain[47]. TRPV1 has an internal diameter of ~6.8 Å[48], which is large enough for QX-314 entry[39]. The pores formed by PFTs are larger than TRPV1 (Hla: 15 Å[34]; leukocidins: 20–30 Å[49]). Future work will determine the exact mechanisms by which QX-314 enter neurons during bacterial infection. Although we have not yet determined these mechanisms, the highly effective and long-lasting silencing of pain by QX-314 is significant in itself.

Pore-forming toxins are major virulence factors for many bacterial pathogens beyond *S. aureus*[50]. It would be interesting to determine whether PFTs contribute to other pathogenic pain mechanisms. Our work demonstrates that length and size of the PFT is not relevant, as both large beta-barrel toxins (Hla, HlgAB) and short amphipathic peptide toxins (PSMs) are capable of

inducing neuronal firing and pain. Given that PFTs are highly damaging agents, an ability for nociceptors to sense their presence could be an important mechanism to warn the host to a pathogen's presence.

We believe there is a significant need to study pain in the context of live infections. Preclinical studies of inflammatory pain often utilize complete Freund's adjuvant or carrageenan, which are not pathophysiological triggers of pain in humans. Injection of bacterial lipopolysaccharides (LPS), flagellin, or zymosan from fungi are more relevant to infection[6,8,47]. However, pathogens produce many virulence factors beyond these components, some that are not readily purified, synthesized, or stored (e.g., heat-labile toxins). Thus, there is a need to better define the molecular mechanisms of pain during live infections, and to determine the effectiveness of analgesics in these settings. Some groups have begun this effort. Klumpp and colleagues showed that pelvic hypersensitivity produced by uropathogenic *E. coli* is dependent on the O-antigen moiety type of LPS and TRPV1[51–53]. Farmer et al.[54] showed that repeated infection of the vaginal tract by *Candida albicans* led to development of robust mechanical allodynia in mice.

In conclusion, our study defines several critical molecular mechanisms of pain during live MRSA bacterial infections. We identify QX-314 as an effective analgesic strategy to silence spontaneous pain, thermal, and mechanical hyperalgesia during infection.

## Methods

**Mice**. C57BL/6 and B6.Trpv1$^{-/-}$ mice were originally purchased from Jackson Laboratories (ME, USA) and animal colonies were maintained in a full barrier specific pathogen free animal facility at Harvard Medical School. Age-matched 6–20-week-old male C57BL/6 mice were used for most spontaneous pain and hyperalgesia experiments in this study. Trpv1$^{+/-}$ heterozygous mice were bred to each other to produce Trpv1$^{+/+}$, Trpv1$^{+/-}$, and Trpv1$^{-/-}$ littermate controls, and both male and females were used for work involving Trpv1. All animal and bacterial experiments were performed following approval by the committee on microbiological safety and Institutional Animal Care and Use Committee (IACUC) at Harvard Medical School. Mouse pictures and video recording were done according to policy as written by the IACUC.

**Statistical analysis**. For analysis of thermal and mechanical hyperalgesia, repeated measures (RM) two-way ANOVA with Tukey's post-test (three or more groups) or Sidak's post-test (two groups) was used to determine statistical significance. For analysis of spontaneous pain, LDH release experiments, tissue swelling, bacterial load, and calcium-imaging experiments, one-way ANOVAs with Tukey's post-test (three or more groups) or unpaired $t$ tests (two groups) were used to determine significance. For in vivo spontaneous pain data sets, data is plotted as mean ± s.e.m. with individual mice represented as individual symbols. For analysis of QX-314 effects on spontaneous pain, data was analyzed using paired $t$ tests. Multi-electrode array experimental results were analyzed with RM one-way ANOVAs with Tukey's post-tests (three or more groups), or paired $t$ tests (two groups). All relevant statistical tests used were two-sided throughout the study. We used Graphpad Prism software (CA, USA) to analyze and plot data. Sample sizes for mouse experiments were powered based on standard numbers in the field. Non-significant (n.s.) was defined as $p > 0.05$.

Mice studies were randomized as appropriate. For infection studies, bacteria were prepared and used to inoculate across cages to ensure equivalent dosages. For transgenic mice studies, littermates were housed together prior to genotyping and studies carried out across multiple cages.

**Bacterial strains and cultures**. *S. aureus* USA300 (LAC), Newman, USA500, and USA300 *S. aureus* isogenic mutant strains lacking all bicomponent leukocidins Δleukocidins (Δ*lukAB* Δ*hlgACB::tet* Δ*lukED::kan* Δ*pvl::spec*), hla (Δ*hla*), and both leukocidins and hla (Δ*lukAB* Δ*hlgACB::tet* Δ*lukED::kan* Δ*pvl::spec* Δ*hla::erm*) were generated by transduction of previously described mutated loci with phage 80a[17,24,45,55,56]. USA300 *S. aureus* isogenic mutants lacking *agr* (Δ*agr*) or the PSM encoding loci PSMα (Δ*psmα*); PSMβ (Δ*psmβ*); Hld (Δ*hld*); and all PSMs (Δ*psmα*Δ*psmβ*Δ*hld*) were described previously[17,24,45,56]. USA300 parental strains are designated as WT throughout the study in comparison with equivalent isogenic mutants. For stationary phase cultures, *S. aureus* was grown overnight (O/N) in Tryptic Soy Broth (TSB, Sigma) at 250 r.p.m., 37 °C (MaxQ 4000, Thermo Scientific). For subsequent mid-log phase growth, a 1:100 dilution of the O/N culture was made into fresh TSB and grown for an additional 3.5 h. The culture was

centrifuged (Sorvall Legend RT, Kendro Lab Products) at 800×g for 15 min and pellet washed once with phosphate-buffered saline (PBS). An A$_{600}$ reading of 0.500 OD (DU 800, Beckman Coulter, Indianapolis, IN, USA) approximated $4 \times 10^8$ CFU per ml; inoculums were prepared based off this. For each experiment, CFU were confirmed on tryptic soy agar (TSA) plates and expected hemolysis verified on TSA with 5% sheep's blood (BD Biosciences). Heat-killed *S. aureus* were made by heating $2.5 \times 10^9$ CFU per ml bacterial suspension (in PBS) in a 100 °C water bath for 15 min. Bacteria were pelleted and resuspended in the same volume of fresh PBS. Heat-killed bacteria were plated to ensure lack of viability.

**Bacterial toxins**. Phenol-soluble modulins PSMα3 (formyl-MEF-VAKLFKFFKDLLGKFLGNN) and δ-toxin (formyl-MAQDIISTIGDLVKWII DTVNFTKK) were synthesized by American Peptide Company (Sunnyvale, CA, USA). They were dissolved in dimethyl sulfoxide (DMSO) to 10 mM, based on peptide content, and stored at −80 °C until use. Vehicle controls for these peptides included appropriate DMSO concentrations. Hla was purchased from Sigma, dissolved in PBS to 1 mg/ml, and stored at −80 °C until use. Recombinant HlgA and HlgB were produced, purified, and assembled into the bicomponent HlgAB as previously described[56,57]. They were used in neuronal and in vivo assays based on the total protein content. For MEA plate experiments, toxins were diluted in neurobasal-A medium (Life Technologies). For animal experiments, toxins were diluted in PBS as a vehicle.

**Treatment of mice and measurements**. For bacterial infections and pain studies, *S. aureus* reconstituted in PBS was injected subcutaneously into the mouse hind paw using a 31 G insulin syringe, 0.5 cc (BD) in a 20 µl volume. Unless otherwise noted, all infections were done using mid-log (exponential) phase bacteria. For measurement of tissue bacterial load, infected paw tissue was excised to the ligaments, weighed, and resuspended in 1 ml of cold PBS. Tissue was dissociated using a Tissue Lyzer II (Qiagen, Hilden, Germany) at 25 s$^{-1}$ for 5 min. Serial dilutions were made, plated, and CFUs counted the next day. Bacterial load was expressed as CFU per mg tissue. For bacterial load measures following spontaneous pain, paw tissues were excised immediately following the end of the pain measurements at the 1 hour time point, and analyzed for bacterial load recovery. For bacterial load measures in QX-314 (lidocaine *n*-ethyl bromide, Sigma)-treated mice, three groups of mice receiving either 1, 2, or 3 treatments (once daily) of PBS or 2% QX-314, before bacterial load was measured, were used. All mice received 20 µl of PBS or 2% QX-314 co-injected with $1 \times 10^6$ CFU of *S. aureus*. At 24 h, the bacterial load of the first treatment group was counted, while the second and third groups received another 20 µl intraplantar injection of 2% QX-314. This process was repeated at 48 and 72 h. Bacterial load was determined as described.

For tissue swelling measurements, hind paws of mice were measured using a digital caliper (Mitutoyo, Aurora, Illinois, USA) both before and after completion of the spontaneous pain assay (1 h). Tissue swelling was calculated as the percentage increase from the baseline paw thickness.

To chemically ablate nociceptor neurons, three increasing doses of RTX (Sigma) —30, 70, 100 µg/kg—were subcutaneously administered in the flank of 4-week-old male B6 mice on consecutive days[8]. Control mice were treated with vehicle (2% DMSO, 0.15% Tween 80 in PBS). Resiniferatoxin or vehicle-treated mice recovered for 4 weeks, and were used for infection studies at 8 weeks of age.

**Behavioral assays**. For spontaneous pain behavior measures, mice were injected into the right hind paw with bacteria or with toxins. The time displaying spontaneous licking, lifting, biting, flinching of injected paw was recorded per min. For measurement of mechanical and heat hyperalgesia, all animals were habituated to the behavioral testing equipment at least three times. Three baseline measurements were taken for each behavioral test. To measure thermal hyperalgesia (heat sensitivity), mice were placed on a glass plate of a Hargreave's apparatus (IITC Life Science, CA, USA) set to 29 °C. A radiant heat source was applied to the dorsal surface of the hind paw and latency measured as the time for the mouse to lift/lick/ withdraw the paw (maximum time of 40 s). For mechanical sensitivity, mice were placed on an elevated wire grid. Von Frey filaments (0.008–6.0 g) were applied to the dorsal surface of the hind paw. A threshold was determined to be the smallest filament producing at least 5 out of 10 responses (lifting, licking, and withdrawing). Observers were blinded to bacterial strain and mouse strain as applicable.

**Multi-electrode array plates**. For neuronal analysis on MEA plates, single-well MEA plates containing 64 electrodes each (Axion BioSystems, Atlanta, GA, USA) were coated with a 5 µl drop of 0.1% Poly(ethyleneimine) in borate buffer (pH 8.4) for 1 h at 37 °C. Plates were rinsed four times with sterile ddH$_2$O and allowed to dry. MEAs were coated in 20 µg/ml laminin (Life Technologies). Dorsal root ganglia from adult B6 mice (7–15 weeks old) were dissected into neurobasal-A medium (Life Technologies) and then dissociated in 1 mg/ml collagenase A and 3 mg/ml dispase II (enzymes, Roche Applied Sciences) in HEPES buffered saline for 60 min at 37 °C. After mechanical trituration, DRG cells were run over a 12% bovine serum albumin (BSA) (Sigma) gradient. The top layers of cellular debris were removed neuronal cells washed, pelleted, and resuspended in B-27 supplemented neurobasal (NB) media containing penicillin/streptomycin (Life Technologies) and 50 ng/ml nerve growth factor. Cells were then dropped at high

density (25,000 cells in a 5 μl droplet) onto the electrodes. One hour later, media containing NGF was added to the cultures. Neurons were maintained on MEAs at 37 °C, 5% $CO_2$ for 7 days, with media changes every 2–3 days. DIV 7-day-old MEA cultures were used for stimulation and analysis.

MEA plates were recorded and data analyzed using the MUSE (Axion BioSystems) system, with the associated Axis computer program (Axion BioSystems). Real-time spontaneous neural configuration with a spike detection criterion of >5.5 STDs was used. The temperature was set to 37 °C and a baseline for each plate recorded before toxin addition. Compounds were added at 10× in NB media. An active electrode was defined as exhibiting ≥5 spikes per min in any 1-min interval after compound addition. Electrodes exhibiting irregular waveforms ("noise") were removed from analysis. For time courses, active electrodes and their corresponding baselines' spike rates were averaged during that minute and plotted (Fig. 3). Average spike rate is the pooled data of the average spike rate over the total time for each active electrode. The number of active electrodes was determined by counting active electrodes per MEA plate and averaging this value over several plates. Well-wide firing rate was calculated per MEA plate by summing the total "spikes" (action potentials) per plate and dividing that by the total time (min). Well-wide firing rate was averaged over several MEA plates. Waveforms and raster plots were generated with NeuroExplorer (Nex Technologies, Madison, AL, USA).

For MEA experiments involving Hla and QX-314, baseline electrode activity (10 min) was taken, followed by 30 μg/ml Hla application (10 min), and finally 5 mM QX-314 added (10 min). For MEA experiments involving PSMα3, a baseline was taken (5 min), followed by 10 μM PSMα3 application (5 min), and finally 5 mM QX-314 (5 min). QX-314 was dissolved in NB media for MEA plate experiments. For dose responses, each dose of toxin was added sequentially, from lowest to highest. Between doses, MEAs were gently washed three to four times in NB media and allowed to recover at 37 °C, 5% $CO_2$, for at least 30 min. Analysis was done as described.

**LDH release assays**. Lactate dehydrogenase (LDH) release was used to assess the viability of DRG neurons using a LDH cytotoxicity assay kit (Cayman Chemical Company, Ann Arbor, MI, USA). DRG neurons were dissected from B6 mice and transferred to laminin-coated 96-well plates (5000 cells per well). On the next day, the media was removed and replaced with Krebs–Ringer solution, and the cells were stimulated with live bacteria, purified toxins, or Triton X (1%, positive control) for 15 min. The plate was centrifuged at 400×$g$ for 5 min and supernatants collected and filtered with a 0.22 μm filter. The filtered supernatants (100 μl) were immediately used for LDH measurement following manufacturer's instructions. Lactate dehydrogenase activity was determined by spectrophotometric absorbance at 490 nm (SpectraMax 340PC, Molecular Devices, Sunnyvale, CA, USA) and data presented as the percentage of total LDH released relative to Triton X group (100% of release).

**Calcium imaging**. DRG neurons were isolated and cultured as described above for MEA plates, except that cells were plated at lower density onto laminin-coated cell culture dishes (no BSA gradient) and the antimitotic inhibitor Ara-C was not added to cultures. DRG neurons were used for calcium imaging within 24 h of plating. For calcium imaging, cells were loaded with 5 μM Fura-2-AM (Life Technologies) at 37 °C for 30 min in neurobasal-A medium, washed into Kreb's Ringer (Boston BioProducts) (KR: 120 mM NaCl, 5 mM KCl, 2 mM $CaCl_2$, 1 mM $MgCl_2$, 25 mM sodium bicarbonate, 5.5 mM HEPES, 1 mM D-glucose, pH 7.2 ± 0.15) and imaged at room temperature. To measure calcium flux in response to bacterial application, *S. aureus* was grown to mid-log phase as described, and resuspended in Kreb's Ringer at 10×, the final concentration in CFU per ml. Bacteria was applied directly to the culture bath, followed by 1 μM capsaicin (Tocris) and 45 mM KCl (Sigma) to activate distinct subsets of neurons. For measurements of neuronal responses to bacterial supernatant, mid-log phase *S. aureus* was pelleted and resuspended as a 10× bacterial suspension ($1.5 \times 10^{10}$ CFU per ml) in Kreb's Ringer buffer. The bacterial suspension was incubated for 3 h at 37 °C with shaking (150 r.p.m.). The bacteria were then centrifuged at 8000 r.p.m. for 5 min and the supernatant collected and stored at −80 °C until use. Calcium imaging was performed using a Nikon Ti-S/L100 inverted microscope (Nikon). Cells were illuminated by an ultraviolet light source (Lambda XL lamp, Sutter Instrument), and fluorescence emission captured by Zyla sCMOS camera (Andor) and 340/380 ratiometric images were processed and analyzed with NIS-elements Advance Research software (Nikon).

**QX-314, lidocaine, and ibuprofen experiments**. QX-314 (lidocaine *n*-ethyl bromide) or lidocaine hydrochloride, and ibuprofen (ibuprofen sodium) were purchased from Sigma-Aldrich. To measure the effects of QX-314 on Hla-induced spontaneous pain, mice were injected with 10 μl of Hla (1 μg), followed by pain behavioral analysis for 15 min. Mice were then injected with either 10 μl of PBS or QX-314 (4%) and behavioral analysis performed for an additional 15 min. To measure the effects of QX-314 and lidocaine on *S. aureus* spontaneous pain, mice were treated with 10 μl of 4% lidocaine, 4% or 8% QX-314, or PBS. Subsequently, mice were infected with 10 μl of $1 \times 10^9$ CFU *S. aureus*, and spontaneous pain assayed as described. Final concentrations for this assay are: 2 or 4% QX-314, 2% lidocaine, $5 \times 10^8$ CFU *S. aureus*. To measure the effects of QX-314 on *S. aureus*-

induced hyperalgesia, $1 \times 10^6$ CFU of *S. aureus* was co-injected with either 2% QX-314 or 2% lidocaine. Mechanical or thermal hyperalgesia was assayed as described above at distinct time points indicated. For experiments where QX-314 was used as a treatment post infection: $1 \times 10^6$ CFU *S. aureus* was injected and mechanical hyperalgesia measured as described. At 23 and 47 h post infection, one measurement was taken followed by treatment with 20 μl of 2% QX-314 (at 24 and 48 h post infection). Measurements were taken at 25 and 48 h post infection and then every 2 h for 6 h on these 2 days. To measure effects of ibuprofen on *S. aureus*-induced pain, 4 mg/kg or 40 mg/kg ibuprofen in PBS was injected intraperitoneally immediately after infection with $1 \times 10^6$ CFU *S. aureus*.

**Data availability**. All relevant data for this paper will be made readily available upon request from the authors.

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

## Acknowledgements

We thank Nicole Lai, Nicole Yang, Kaitlin Goldstein, Rianne Haarsma, Yan Zhou, Pingchuan Ma, Amélie Bouvier, Lin Li, Yichao Shen, and Talia Kirshbaum for technical help. We thank Amer Villaruz for generously providing *S. aureus* isogenic mutant strains, recombinant toxins, and experimental advice. We thank Juliane Bubeck-Wardenburg and Georgia Sampedro for advice on *S. aureus* skin infections. We thank Bruce Bean for mentoring and advice on QX-314, Simon Dove and Gerald Pier for mentoring, Dave and Lynnae Blake for moral support. This work was generously supported by funding from the National Institutes of Health (NIH) under grant numbers NCCIH DP2AT009499 (I.M.C.), NIAID K22AI114810 (I.M.C.), NIGMS T32 GM007308 (A.L.), NIAID T32 AI007180 (A.L.), F30 AI124606 (A.L.), NIAID R01 AI105129 (V.J.T.), R01 AI099394 (V.J.T.), R01 AI121244 (V.J.T.), Intramural Research Program of the NIAID, NIH ZIA AI000904 (M.O.), NINDS NS039518 (C.J.W.), the DoD W81XWH-15-1-0480 (C.J.W.), and by a Kaneb fellowship award (I.M.C.). V.J.T. is a Burroughs Wellcome Fund Investigator in the Pathogenesis of Infectious Diseases.

## Author contributions

K.J.B. and I.M.C. planned and designed all the experiments; K.J.B. coordinated all the experiments. K.J.B., T.V., P.B., F.A.P.-R., K.L.A. and Y.C.M. performed pain behavioral experiments. T.V. and P.B. performed calcium-imaging experiments; K.J.B. and K.L.A. carried out MEA studies; F.R. performed LDH release assay; M.O., V.J.T. and A.L. generated and provided bacterial strains and recombinant toxins. C.J.W. and D.P.R. provided knowledge on QX-314 treatment; K.J.B. and I.M.C. wrote and edited the manuscript.

## Additional information

**Competing interests:** The authors declare no competing financial interests.



