## [Peer Review File · Nature Communications]

Reviewers' comments:

Reviewer #1 (Remarks to the Author):

Infections by bacteria or fungi are often accompanied by pain at the site of infection. The mechanisms through which the presence of pathogen is translated into painful signals remain poorly understood.

Pain is initiated when peripheral nociceptors, the pain-transducing somatosensory neurons, are excited by a noxious stimulus. While bacterial products such as lipopolysaccharide can induce pain in animal models, it is unknown which of these products elicit pain in vivo during an active infection.

In the present manuscript the authors seek to identify the pain-inducing bacterial factors from *Staphylococcus aureus*, a common human pathogen that causes painful infections. Initially the authors inject cultured and washed live bacteria of a MRSA strain, a variety resistant even against last resort antibiotics, into the mouse paw and monitored pain behavior over 60 minutes. At the highest bacterial density, mice showed clear signs of pain behavior. Moreover, while acute pain was transient, mice developed thermal and mechanical hyperalgesia, a painful hypersensitivity to these stimuli.

Bacteria co-cultured with dorsal root ganglia neurons elicited calcium influx. When co-cultured with a strain lacking virulence genes (*agr* locus) encoding for soluble bacterial toxins, sensory neurons failed to respond. Bacterial supernatants, likely containing these toxins, were sufficient to activate neurons. The authors observed that all three toxin classes encoded by the locus excited cultured neurons and induce pain behavior when injected in purified form. However, when mice were infected with strains lacking specific toxins, only bacteria lacking alpha-hemolysin failed to induce acute pain, suggesting that this pore forming toxin acts on sensory neurons in vivo.

Interestingly, alpha-hemolysin-deficient bacteria still elicited thermal and mechanical hyperalgesia, the former depending on TRPV1, the capsaicin receptor and heat sensor. TRPV1 is permeable to large ions, a potential route to introduce therapeutic into the neuronal cell. In the present manuscript the authors administer QX-314, a TRPV1-permeable analgesic, observing clear analgesic effects.

This is a quite elegant study with large datasets and supporting information. The experiments are presented clearly, with no major flaws to be found. The use of a live infection model is clearly an advance over prior studies using purified bacterial products. However, this model requires further discussion. Do the authors think that the injection of a quite large number of bacteria in log phase resembles an early infection in a human? Human infections usually start with exposure to smaller amounts of bacteria that increase in number more slowly until pain is sensed. Why does acute pain behavior subside quickly ~30 min after injection? Does it mean the bacteria suddenly stop making hemolysin? Or are they attacked by the immune system? Since hyperalgesia persists after the acute pain phase, how is it maintained? Is it a sign of a beginning immune response, potentially neutrophils or macrophages infiltrating the tissue and releasing factors that sensitize nociceptors, such as proteases, oxidative mediators etc..

Also, an introduction of mouse vs. human *S. aureus* infections would be helpful, especially for readers in the pain field who might not have sufficient insight into microbiology and infectious disease models. Do mice always clear infections with human *S. aureus* without antibiotic support? How does this compare to a human with a chronic painful MRSA skin infection?

Minor:

Lines 218 etc.: Authors use the term "this data" here and at several other positions in the text. Maybe use "these data", or "the data" instead? Copy editor can decide.

Lines 507/8: "LPS has also been shown to directly mechanically gate TRPA1 channels"; Please remove the word "mechanically", since it is unclear how exactly LPS gates TRPA1.

Line 543: "physiological", maybe replace with "pathophysiological"?

Reviewer #2 (Remarks to the Author):

This manuscript concerns mechanisms by which infection by *Staph aureus* bacteria produces sensory neuron activation (and consequently pain). The authors show that the pore-forming toxin, alpha-hemolysin, is the main culprit in activating nociceptors, which results in acute pain, as well as thermal and mechanical hypersensitivity. Pharmacological and genetic evidence supports the critical involvement of the TRPV1 ion channel in thermal hypersensitivity, consistent with its known function in heat detection. Furthermore, the authors show that administration of the sodium channel blocker, QX-314, diminishes alpha-hemolysin-evoked pain, possibly as a consequence of its entry through alpha-hemolysin-generated pores.

Overall, this study is thorough and convincing. The authors have used a variety of methods - cellular and behavioral - to address their questions concerning *Staph aureus* actions on nociceptors. The main issue is one of novelty since the main and most significant conclusion of this study has already been described by this group (Chiu et al, Nature 2013). In this previous study, the authors show that *Staph aureus* activates nociceptors through release of alpha-hemolysin. The demonstration here that thermal hypersensitivity requires TRPV1 is not especially new since this has been demonstrated for all manner of noxious and inflammatory agents. Furthermore, the use of QX-type anesthetics to silence nociceptors via entry through non-selective cation channels has also been extensively described by this group (Binstok et al. Nature 2007). In the current study, the authors suggest that QX-314 enters neurons through the alpha-hemolysin pore, but this is not rigorously explored.

In summary, this study is well executed but provides only an extension of previous observations by these authors. Consequently, the study seems better suited to a more specialized journal.

Point by point response:

We thank the reviewers for their thorough review of our manuscript. Below we discuss and highlight changes to the manuscript that we have made to address the concerns of the referees. For reviewer #2, we address the conceptual and therapeutic novelties and advances of our paper. We believe our manuscript is significantly strengthened and hope it is acceptable to the reviewers and editors.

We have underlined all revised changes made in the manuscript.

Referees' comments

Reviewer #1 (Remarks to the Author):

Infections by bacteria or fungi are often accompanied by pain at the site of infection. The mechanisms through which the presence of pathogen is translated into painful signals remain poorly understood.

Pain is initiated when peripheral nociceptors, the pain-transducing somatosensory neurons, are excited by a noxious stimulus. While bacterial products such as lipopolysaccharide can induce pain in animal models, it is unknown which of these products elicit pain in vivo during an active infection.

In the present manuscript, the authors seek to identify the pain-inducing bacterial factors from *Staphylococcus aureus*, a common human pathogen that causes painful infections. Initially the authors inject cultured and washed live bacteria of a MRSA strain, a variety resistant even against last resort antibiotics, into the mouse paw and monitored pain behavior over 60 minutes. At the highest bacterial density, mice showed clear signs of pain behavior. Moreover, while acute pain was transient, mice developed thermal and mechanical hyperalgesia, a painful hypersensitivity to these stimuli.

Bacteria co-cultured with dorsal root ganglia neurons elicited calcium influx. When co-cultured with a strain lacking virulence genes (*agr* locus) encoding for soluble bacterial toxins, sensory neurons failed to respond. Bacterial supernatants, likely containing these toxins, were sufficient to activate neurons. The authors observed that all three toxin classes encoded by the locus excited cultured neurons and induce pain behavior when injected in purified form. However, when mice were infected with strains lacking specific toxins, only bacteria lacking alpha-hemolysin failed to induce acute pain, suggesting that this pore forming toxin acts on sensory neurons in vivo. Interestingly, alpha-hemolysin-deficient bacteria still elicited thermal and mechanical hyperalgesia, the former depending on TRPV1, the capsaicin receptor and heat sensor. TRPV1 is permeable to large ions, a potential route to introduce therapeutic into the neuronal cell. In the present manuscript the authors administer QX-314, a TRPV1-permeable analgesic, observing clear analgesic effects.

This is a quite elegant study with large datasets and supporting information. The experiments are presented clearly, with no major flaws to be found. The use of a live infection model is clearly an advance over prior studies using purified *bacterial products*. *However, this model requires further discussion.*

Responses to Reviewer #1

1) *Do the authors think that the injection of a quite large number of bacteria in log phase resembles an early infection in a human? Human infections usually start with exposure to smaller amounts of bacteria that increase in number more slowly until pain is sensed.*

- A major reason that we chose the dosages of bacteria that we used in our study is based on other work in the field of MRSA pathogenesis. Other studies using *S. aureus* in mouse models of subcutaneous skin infections need 10^7 to 10^9 CFU to establish infection. Since the strains of *S. aureus* used in this study (USA300, USA500, Newman) are human clinical isolates, and not mouse adapted strains, large amounts of bacteria are typically needed to induce infection. This highlights the differences in mouse and human host responses to *S. aureus* infection as pointed out by the reviewer, including the lack of the c5aR receptor (receptor for PVL, a leukocidin) in mice and irrelevance of several *S. aureus* factors that neutralize human innate defenses in mouse models. Despite these differences, mice are a widely used model to study *S. aureus* pathogenesis and virulence in a variety of infection contexts. Therefore, we believe our findings on pain mechanisms in mice could translate into human pain mechanisms.
- We do not think an injection of a large number of bacteria resembles an early infection in a human. Rather, the doses we used to produce spontaneous pain reflexes, especially in our immediate post-infection pain assays, likely mimic those at peak infection where this type of pain would be most prominent. In addition, the dose we chose falls within the range of dosages (in CFUs) commonly used in USA300 skin infection models. We believe the spontaneous pain reflexes we observed (lifting/licking/flinching of the hind paw) may reflect sporadic, stabbing pain that occurs in humans at the peak of infection.
- In the results, we have now added a description of spontaneous pain reflexes and the dosages of bacteria in relation to other studies (page 6, lines 144-148).
- We have also added two paragraphs about the differences between mouse models of and human *S. aureus* infections and what spontaneous pain behaviors we observed could implicate in humans in the discussion (pages 20-22, lines 457-491).

2) *Why does acute pain behavior subside quickly ~30 min after injection? Does it mean the bacteria suddenly stop making hemolysin? Or are they attacked by the immune system?*

- The time course of analysis was over 1 hour, but we do not think that this behavior goes away after 30 minutes only (see Supplemental Figure 1a). The decrease in the spontaneous pain behavior likely occurs due to mice becoming inactive during the assay time, but not because they are no longer experiencing pain (they fall asleep). Individual mice will wake up and display spontaneous pain behaviors, but across the cohort of mice the pain behaviors observed are more sporadic than at the beginning

when the mice are all awake at the same time. We believe this continues throughout the day as long as the mice are active and awake.

- We should have qualified that the behavior we observed immediately following infection was not acute pain, meaning only occurring in the beginning of infection, but rather “spontaneous” reflexive nocifensive pain behaviors that occur throughout infection. We now use the term “spontaneous” throughout the manuscript.
- We believe spontaneous pain reflexes may reflect the stabbing, sporadic pain patients may feel throughout the time course of severe local infections. Of note, while many substances produce hyperalgesia, or increased pain sensitivity, a noxious stimulus has to be especially strong or intense to produce spontaneous pain reflexes. Capsaicin or mustard oil, for example, produces spontaneous overt pain. This indicates that live bacteria, but not dead bacteria, are capable of producing very intense pain that induces spontaneous pain reflexes.
- To clarify this, we have now replaced the term “acute pain” with “spontaneous pain” or “spontaneous pain reflexes” throughout the figures and manuscript, and addressed this point in the introduction and in the discussion.
- In the introduction, two sentences on spontaneous pain reflexes were added (page 3, lines 70-72, 75-77).
- In the results, spontaneous pain and the relevance of the bacterial dosages used are now described (page 6, lines 144-148).
- A paragraph has been added to the discussion to describe spontaneous pain and its implications (page 21, lines 474-491).

3) Since hyperalgesia persists after the acute pain phase, how is it maintained? Is it a sign of a beginning immune response, potentially neutrophils or macrophages infiltrating the tissue and releasing factors that sensitize nociceptors, such as proteases, oxidative mediators etc.

This is a good question, particularly since TRPV1 mediates heat hyperalgesia, whereas *agr* did not. *agr* mediated spontaneous pain reflexes, whereas TRPV1 did not, indicating there are likely other factors besides bacteria-derived mechanisms that lead to hyperalgesia during infection.

- There could be several factors mediating TRPV1 sensitization: one mechanism could be the drop in pH that occurs during infection as protons are one of the known ligands for TRPV1. This is now discussed on page 23, lines 518-521.
- The other mechanisms of TRPV1 sensitization could be through what the reviewer mentions, which are infiltrating immune cells such as neutrophils, macrophages, and

their release of inflammatory mediators such as proteases, oxidative mediators, and cytokines.

- Other neuronal mechanisms (ion channels, cytokine receptors) or even bacterial products not considered in this manuscript could also lead to hyperalgesia.
- We have now added a paragraph in the discussion about these potential mechanisms of maintaining hyperalgesia (page 23, lines 518-529).
- As described above, spontaneous pain occurs throughout the infection. We designed our spontaneous pain assay for ease of measurement and to ensure the mice were naïve to spontaneous pain (they will sleep or hide “protect” the paw in an attempt to mitigate this pain after time, so the response seen in our assay during the first 30 minutes is the strongest).

4) Also, an introduction of mouse vs. human S. aureus infections would be helpful, especially for readers in the pain field who might not have sufficient insight into microbiology and infectious disease models. Do mice always clear infections with human S. aureus without antibiotic support? How does this compare to a human with a chronic painful MRSA skin infection?

- This is a great point and related to point #1. We have added a paragraph on mouse models of *S. aureus* infection vs. human infection in the discussion section (page 20-21, lines 457-473). Clearance of human *S. aureus* without antibiotic support is dependent on the dose of bacteria and the type of infection (skin, bacteremia, etc.). The type of infection modeled here would be considered a short-term (not recurrent) *S. aureus* infection, which is a very common type of infection; our model is representative of a fulminant invasive skin/soft tissue infection that is cleared over time. Clearance of MRSA is seen in other mouse models of subcutaneous skin infections (such as in abscess models to study *S. aureus* virulence).
- One example of a chronic MRSA infection could be atopic dermatitis – where over 90% of people with this skin condition are colonized with the bacterium. This would elicit itch sensations, as the epidermis is primarily innervated by itch, not pain, mediating sensory neurons. We have a new study in the lab which is specifically focused on studying whether itch is also produced by *S. aureus* longer term infections, but it is beyond the scope of this current study.

Minor:

6) Lines 218 etc.: Authors use the term “this data” here and at several other positions in the text. Maye use “these data”, or “the data” instead? Copy editor can decide.

We thank the reviewer for bringing up this point. We have now changed this term to “these data”.

7) Lines 507/8: “LPS has also been shown to directly mechanically gate TRPA1 channels”; Please remove the word “mechanically”, since it is unclear how exactly LPS gates TRPA1.

We have removed the term mechanically and agree that it is unclear how LPS gates TRPA1.

8) Line 543: “physiological”, maybe replace with “pathophysiological”?

We appreciate the reviewer finding this difference. We have now replaced physiological with “pathophysiological” on page 24, line 556.

Reviewer #2 (Remarks to the Author):

This manuscript concerns mechanisms by which infection by *Staph aureus* bacteria produces sensory neuron activation (and consequently pain). The authors show that the pore-forming toxin, alpha-hemolysin, is the main culprit in activating nociceptors, which results in acute pain, as well as thermal and mechanical hypersensitivity. Pharmacological and genetic evidence supports the critical involvement of the TRPV1 ion channel in thermal hypersensitivity, consistent with its known function in heat detection. Furthermore, the authors show that administration of the sodium channel blocker, QX-314, diminishes alpha-hemolysin-evoked pain, possibly as a consequence of its entry through alpha-hemolysin-generated pores. Overall, this study is thorough and convincing. The authors have used a variety of methods - cellular and behavioral - to address their questions concerning *Staph aureus* actions on nociceptors. The main issue is one of novelty since the main and most significant conclusion of this study has already been described by this group (Chiu et al, Nature 2013). In this previous study, the authors show that *Staph aureus* activates nociceptors through release of alpha-hemolysin. The demonstration here that thermal hypersensitivity requires TRPV1 is not especially new since this has been demonstrated for all manner of noxious and inflammatory agents. Furthermore, the use of QX-type anesthetics to silence nociceptors via entry through non-selective cation channels has also been extensively described by this group (Binshtok et al. Nature 2007). In the current study, the authors suggest that QX-314 enters neurons through the alpha-hemolysin pore, but this is not rigorously explored. In summary, this study is well executed but provides only an extension of previous observations by these authors. Consequently, the study seems better suited to a more specialized journal.

Response to Reviewer #2:

We appreciate the time that reviewer #2 spent reading our manuscript. Though they appreciated the thorough and convincing execution, their main concern was regarding the novelty of our study in relation to our previous work. We strongly believe that there are several major advances in our current study that was not previously shown in our own work (Chiu et al, Nature 2013) or by other groups.

Below we discuss specific points that make our study an important advance for the understanding of pain mechanisms during live bacterial infections, and for the

therapeutic treatment of pain during infections. We also have extensively revised our manuscript to point out these advances in the introduction and discussion sections of our manuscript (pages 3-6 and 20-25).

- 1) The first major novelty was microbiological. We found that three distinct, major classes of bacterial pore forming toxins: 1) small peptide toxins: phenol soluble modulins (PSMs), 2) the beta-barrel toxin: α -hemolysin (Hla), and 3) the bicomponent leukocidin toxin: HlgAB can all activate nociceptor neurons, produce ionic flux in neurons, cause neurons to fire action potentials, and produce spontaneous pain in mice. This newfound general mechanism of pore forming toxin induced pain has never been shown before and implies that any type of bacterial pore forming toxin, of which there are thousands amongst many diverse bacteria, could directly activate neurons by forming membrane pores to allow ionic influx to produce pain.
 - a. Introduction: page 5, lines 103-126
 - b. Discussion: page 24, lines 546-553

- 2) The second novelty is finding a direct correlation between live bacterial infections, virulence, and spontaneous pain. The spontaneous pain assay has never been used before with live pathogens until this manuscript. We believe this assay could be used with a diverse number of invasive bacteria to determine mechanisms of spontaneous pain during infection. In this study, we used three different clinical isolates of *Staphylococcus aureus* (USA300, USA500, and Newman) to show that each strain leads to differing amounts of pain, likely dependent on toxin expression. Pain has not been previously compared between strains of the same bacterium. Of note, highly invasive bacterial infections such as necrotizing fasciitis are characterized by “pain out of proportion” with other physical symptoms early on during infection. Our work indicates that the more virulent the pathogen, the more painful the infection, and so pain could be an indicator/biomarker for bacterial invasion.
 - a. Introduction: page 4-5, lines 83-87, 108-110, 121-122
 - b. Discussion: page 24-25, lines 554-569

- 3) The third novelty is finding a novel role for the quorum sensing component, *agr*, in both neuronal activation and pain production. *agr* is turned on during conditions of high bacterial density and stress within the host, which increases the pathogen’s virulence in its environment. Therefore, pain may be a reflection of bacterial virulence. This is related to point #2. *agr* is known to be one of the major contributors to MRSA’s virulence. Notably, *agr* also critically controls both neuronal calcium influx and spontaneous pain production *in vivo* in our infection model. Thus, pain could be used as a readout for virulent and invasive bacterial infections and as a biomarker for its treatment.
 - a. Introduction: page 5, lines 116-126
 - b. Discussion: page 22, lines 492-495

- 4) The fourth novelty, is that for the first time we determined a key role for the neuronal

ion channel, TRPV1, in thermal hyperalgesia during live pathogenic infection. *agr* isogenic mutants had no effect on hyperalgesia during infection, thus showing pain during infection is not solely mediated by pore-forming toxins. This mechanism is a new finding for *S. aureus* induced pain during infection. These results show that TRPV1 mediated hyperalgesia likely is a mechanism that extends to other *S. aureus* strains not expressing *agr*. This is key as QX-314 (as discussed below), can be delivered via TRPV1 channels, and thus pain (related to hyperalgesia) likely can be treated in pore-forming toxin deficient strains of *S. aureus*, underscoring the importance of this finding.

- a. Introduction: page 6, lines 124-126
- b. Discussion: page 23, lines 514-521

- 5) The fifth novelty, and potentially the most important contribution and advance of our study is finding a highly effective therapeutic way to block pain during infection without affecting host defense. This treatment was using the sodium channel blocker, QX-314. MRSA infections are becoming increasingly significant and understanding how these antibiotic-resistant bacteria interact with the host is of utmost importance. Targeting these interactions could lead to clinical improvement of life for patients. Painful bacterial infections are a major issue in the dentist's office. Lidocaine is known to be neutralized by pH and inflammation, and we show it lasts for a limited time in bacterial infected tissues. Moreover, we find that ibuprofen has no effect on pain. Our study shows that QX-314, by contrast, has hours of analgesic effects. We believe this could be applied translationally in clinical treatment of infections. For example, dentists often encounter painful dental carries and infectious disease doctors encounter infected skin abscesses. QX-314 may be utilized in silencing these forms of infectious pain much more effectively than lidocaine or ibuprofen.
- 6) Though QX-314 was first introduced as a concept in 2007 related to ion channels, it has never been shown to work through delivery via pore-forming toxins. We show that QX-314 can be delivered into neurons to silence firing and pain produced by the bacterial pore-forming toxins Hla and PSM α 3. Given that we believe many bacterial pore-forming toxins would relate to pain not only in MRSA infections but other bacterial infections, QX-314 could be broadly applicable in these settings. Moreover, we could imagine repurposing pore-forming molecules to deliver this analgesic, particularly if they are targeted towards specific subtypes of neurons.

Points 5-6 on QX-314:

Introduction: page 6, lines 127-135

Discussion: page 20-25, lines 444-456, 530-545, 570-575

REVIEWERS' COMMENTS:

Reviewer #1 (Remarks to the Author):

I thank the authors for addressing my concerns. The manuscript is very much improved. The information and discussion provided about the use of the mouse as an infection model, about species differences between mice and humans, and the response of the mouse to the human pathogens used in the study are very helpful and provide perspective. With some limitations, the authors make a good case for the mouse displaying infection-induced pain behavior reminiscent to pain during the peak infection in humans.

Reviewer #3 (Remarks to the Author):

This is an interesting article which refines findings presented by the authors in a 2013 publication (PMID: 23965627) describing pain activation by *S. aureus* and the secreted toxin, alpha-hemolysin (Hla). In the current manuscript, the authors refine the timeframe of spontaneous pain induction, the role of TRPV1 in this response, provide additional information on the role of Hla in pain, along with HlgAB and the PSMs, and demonstrate use of QX-314 as a pain blocker. The authors expand on their previous report using an Hla deletion mutant to include an agr deletion mutant, which in addition to not producing Hla, is also defective in producing other secreted toxins. Overall, the findings reported are a reasonable expansion on the previous findings.

More specific concerns are as follows:

- 1) In 2014, Berube et al (PMID: 24866799) reported that the psm-alpha locus regulates Hla production during infection (using psm-alpha specific or general psm/hld deletion mutants). This group also demonstrated impaired Hla production in vivo with these mutants. Therefore, given the emphasis on Hla as a pain inducer, it is surprising that in the current study, the authors only saw a trend toward decreased pain with the psm mutants. This discrepancy should be thoroughly addressed in the discussion.
- 2) The authors provide no evidence that QX-314 is delivered through toxin pores. All such claims should be removed from the manuscript.
- 3) There are multiple references missing throughout the manuscript. Pg 21, lines 483-484. Please provide references and examples of mice sleeping or hiding paws as a means of pain avoidance. Pg 21, lines 474-477, please provide references for the pain felt by humans during local *S. aureus* skin infections. Pg 22, lines 505-506, provide references for induction of autophagy and other repair mechanisms in response to PFTs. Also, please provide references for agr-regulation of each of the individual toxins used in this study.
- 4) Toxin concentrations should be reported in both ug/mL and molarity to allow direct comparisons.

Point by point response to Referee's Comments

We thank the editor and reviewers for their thorough review of our manuscript. Below we supply a point by point response for each concern, and discuss revisions to the manuscript that we have made. For the reviewer's comments, we have also underlined all changes made in the manuscript.

Reviewer #3 Points:

We thank reviewer #3 for their comments and suggestions. We have addressed each concern below.

This is an interesting article which refines findings presented by the authors in a 2013 publication (PMID: 23965627) describing pain activation by S. aureus and the secreted toxin, alpha-hemolysin (Hla). In the current manuscript, the authors refine the timeframe of spontaneous pain induction, the role of TRPV1 in this response, provide additional information on the role of Hla in pain, along with HlgAB and the PSMs, and demonstrate use of QX-314 as a pain blocker. The authors expand on their previous report using an Hla deletion mutant to include an agr deletion mutant, which in addition to not producing Hla, is also defective in producing other secreted toxins. Overall, the findings reported are a reasonable expansion on the previous findings.

More specific concerns are as follows:

- 1) *In 2014, Berube et al (PMID: 24866799) reported that the psm-alpha locus regulates Hla production during infection (using psm-alpha specific or general psm/hld deletion mutants). This group also demonstrated impaired Hla production in vivo with these mutants. Therefore, given the emphasis on Hla as a pain inducer, it is surprising that in the current study, the authors only saw a trend toward decreased pain with the psm mutants. This discrepancy should be thoroughly addressed in the discussion.*

Response: We appreciate this reference and have now addressed it in our manuscript with a paragraph about this paper:

- *Berube et al* showed that USA300 PSM mutant strains produced significantly less Hla than WT strains at the 3 hr time point in their cultures. Of note, *Berube et al* did find that Hla production by PSM mutant strains was restored to levels of WT bacteria by the 6 hr time point in their study, indicating that there are likely other levels of regulation beyond PSM α . Nonetheless, this paper suggests that the trend toward decreased pain that we found in PSM mutants could be reflective of changes in Hla expression downstream of PSMs rather than the loss of PSMs themselves. We have now written about this possibility in the Discussion of our manuscript.
- Another thought to consider is the differing growth conditions between their study and ours. *Berube et al* diluted an overnight (O/N) culture 1:250 and then took measurements of Hla at 3 and 6 hrs. We diluted an O/N culture 1:100, which is a higher starting concentration, and used a 3.5-hr culture for infection. Therefore, we likely have more Hla production in the PSM mutant strains compared to those at their 3 hr time point (shown in their Figure 2b). Since they observed that full production of Hla is restored by 6 hrs – it is possible that our growth conditions correspond more to between the 3 hr and 6 hr time points in the paper and could explain why we did not see a significant difference in spontaneous pain in PSM mutants whereas we did for Hla mutant bacteria.
- We have now included a paragraph about this paper and its relation to our work in our Discussion on **Page 20, Lines 1201-1208**, and included it as a new reference (**46**).

2) *The authors provide no evidence that QX-314 is delivered through toxin pores. All such claims should be removed from the manuscript.*

Response: We thank the reviewer for bringing up this concern and it matches previous concerns by reviewer #2. We have now worked diligently to remove all claims that QX-314 is delivered through toxin pores throughout our manuscript. To address this concern:

- We have removed **Figure 8b** and its corresponding legend, which originally showed that QX-314 is delivered through pore-forming toxins.
- We have removed all claims throughout the text of the manuscript that stated that QX-314 is delivered through pore-forming toxins that were originally on **Pg. 2, line 61; Pg. 15, lines 865; pg. 17, lines 967; pg. 21, lines 1445-1449**.

- 3) There are multiple references missing throughout the manuscript. Pg 21, lines 483-484. Please provide references and examples of mice sleeping or hiding paws as a means of pain avoidance. Pg 21, lines 474-477, please provide references for the pain felt by humans during local *S. aureus* skin infections. Pg 22, lines 505-506, provide references for induction of autophagy and other repair mechanisms in response to PFTs. Also, please provide references for agr-regulation of each of the individual toxins used in this study.

Response: Thank you for these observations. We have included the following references for each point that was raised.

- **Mice sleeping or hiding paws as a means of pain avoidance:** References **41** (Leys LJ, et al. *Pain*, 2013), **42** (Cortright DN et al. *Expert opinion Drug Discov*, 2008): pg. 19, line 1131.
- **Pain felt by humans during *S. aureus* infection:** References **4** (Chiu IM et al. *Pain* 2016), **14** (Lowy FD N. *Engl J Med*, 1998): pg. 19, lines 1125.
- **Autophagy and other repair mechanism in response to PFTs:** **45** (Maurer K et. al *Cell Host Microbe*, 2015): pg. 20 line 1195.
- **agr-regulation of each individual toxins:** References **18** (Bronner S et al. *FEMS Microbiol Rev*, 2004), **19** (Cheung GY et al. *Infect Immun* 2011), **20** (Novick RP et al. *Ann Rev Gen*, 2008): pg. 20, lines 1187.

4) Toxin concentrations should be reported in both $\mu\text{g/mL}$ and molarity to allow direct comparisons.

Response: Thank you for this point. We have now addressed this concern and included toxin concentrations in both units ($\mu\text{g/mL}$ and molarity) so that comparison between the toxins can be made for each similar experiment.

- Main Figures 3 and 6 were updated to include both toxin concentrations in the figure legends.
- Supplemental Figures 4-6 were updated to include both toxin concentrations in the figure legends.